# Study on the Spatiotemporal Evolution and Influencing Factors of Agricultural Carbon Emissions in the Counties of Zhejiang Province

**DOI:** 10.3390/ijerph20010189

**Published:** 2022-12-23

**Authors:** Changcun Wen, Jiaru Zheng, Bao Hu, Qingning Lin

**Affiliations:** 1Institute of Rural Development, Zhejiang Academy of Agricultural Sciences, Hangzhou 310021, China; 2College of Economics and Management, Zhejiang A&F University, Hangzhou 311300, China; 3Institute of Agricultural Economics and Development, Chinese Academy of Agricultural Sciences, Beijing 100081, China

**Keywords:** Zhejiang Province, agricultural carbon emissions, county territory, spatial spillover effects, influence factor

## Abstract

The accurate measurement of agricultural carbon emissions and the analysis of the key influential factors and spatial effects are the premise of the rational formulation of agricultural emission reduction policies and the promotion of the regional coordinated governance of reductions in agricultural carbon emissions. In this paper, a spatial autocorrelation model and spatial Dubin model are used to explore the spatiotemporal characteristics, influential factors and spatial effects of agricultural carbon emissions (ACEs). The results show that (1) From 2014 to 2019, the overall carbon emissions of Zhejiang Province showed a downward trend, while the agricultural carbon emission density showed an upward trend. ACEs are mainly caused by rice planting and land management, accounting for 59.08% and 26.17% of the total agricultural carbon emissions, respectively. (2) The ACEs in Zhejiang Province have an obvious spatial autocorrelation. The spatial clustering characteristics of the ACEs are enhanced, and the “H-H” cluster is mainly concentrated in the northeast of Zhejiang, while the “L-L” cluster is concentrated in the southwest. (3) The results of the Dubin model analysis across the whole sample area show that the ACEs exhibit a significant spatial spillover effect. The disposable income per capita in the rural areas of the county significantly promotes the increase in the ACEs in the neighboring counties, and the adjustment of the industrial structure of the county has a positive effect on the agricultural carbon emission reductions in neighboring counties. (4) The grouping results show that there is heterogeneity between 26 counties in the mountainous areas and non-mountainous areas. In the 26 mountainous counties, the urbanization rate, rural population, mechanization level and industrial structure have significant negative spatial spillover effects on the carbon emissions. In the non-mountainous counties, the agricultural economic development level and disposable income per capita of the rural residents have significant spatial spillover effects on the agricultural carbon emissions. These research results can provide a theoretical basis for the promotion of the development of low-carbon agriculture in Zhejiang according to the region and category.

## 1. Introduction

Global warming, caused by the greenhouse effect, poses a great challenge to the sustainable development of human society [1]. In particular, greenhouse gases generated by human activities are the main cause of climate change [2]. Carbon emission reduction has become an important topic related to human survival and development. In 2021, under the goals of carbon peak and carbon neutralization, the Chinese government issued the Fourteenth Five Year Plan for the Development of Green Agriculture [3], aiming to control the input of chemical fertilizers, pesticides and other sources of pollution. Agriculture is an important factor in global warming [4]. In total, 13.5% of global greenhouse gases and 53% of global non-CO_2_ emissions stem from the agricultural sector [5], while 17% of China’s greenhouse gases, 50% of CH_4_ and 92% of N_2_O stem from the agricultural sector [6]. Therefore, reducing agricultural carbon emissions (ACEs) is of great significance for the mitigation of climate change and for the achievement of the “double carbon” goal [7,8]. As the first provincial pilot demonstration area for the promotion of sustainable agricultural development and the first pilot area for green agricultural development in China, Zhejiang Province plays an important role as a model for the process of sustainable agricultural development. Research on the realistic characteristics of, and factors influencing, ACESs in Zhejiang Province has important practical value and significance as a reference for the scientific formulation of regional agricultural carbon emission reduction policies.

The current research on ACESs mainly focuses on the measurement of agricultural carbon emissions, the mechanism of agricultural carbon emission reduction and policy research conducted to achieve it, and the factors influencing agricultural carbon emissions. Firstly, the accurate measurement of ACESs is the basis of the relevant research. Compared with industrial carbon emissions, ACESs are more diverse and measured using a variety of methods, including the IPCC emission factor method, model method and life cycle method, which have been used by scholars to measure agricultural carbon emissions. This research includes the assessment of agricultural carbon emission performance [9], the agricultural carbon footprint [10,11], agricultural carbon emission intensity [12] and agricultural carbon compensation [13]. In summary, the existing studies on ACESs mainly focus on the national, regional and micro-levels. For example, Berdanier and Conant (2012) [14] measured ACESs in different countries. Tian et al. (2014) [15] used the IPCC emission factor method to measure ACESs in 31 provinces of China and found significant regional differences in the ACESs across China. Beauchemin et al. (2010) [16] measured carbon emissions from Canadian pastures using the life cycle method and found that intestinal CH_4_ was the main source of carbon emissions arising from pastures. The different methods of measuring ACESs have their own advantages and limitations. The IPCC method has the widest range of applications and is favored by many scholars [5,8] but lacks the capacity to account for regional differences. The life cycle method can measure carbon emissions from the perspective of the whole cycle, but its operation is not feasible, and there are many subjective factors. As a result, the results of different methods are variable. For example, Huang Zuhui and Mi Songhua (2011) [17] calculated the agricultural carbon footprint of Zhejiang Province using the layered input–output life cycle method and compared the results with the ACESs calculated using the Intergovernmental Panel on Climate Change (IPCC) method. Results the ACESs calculated using the IPCC method accounted for only 43.55% of the actual total agricultural carbon footprint. Therefore, when calculating agricultural carbon emissions, local agricultural practices should be combined in order to select the appropriate emission coefficient.

Secondly, research has focused on the mechanisms and policies of agricultural carbon reduction. In terms of agricultural carbon emission reduction policies, the authors of the studies believe that carbon emission trading pilot policies [18], reasonable agricultural subsidy policies [8] and carbon sinks [19] can effectively reduce ACESs and improve the capacity for sustainable agricultural production. In terms of the mechanism of agricultural carbon emission reduction, the authors of the studies believe that green production methods, such as the deep tillage and loose soil methods [20], no-tillage sowing [10] and straw returning [21] can effectively reduce agricultural carbon emissions. The transformation and upgrading of agricultural machinery to improve its energy efficiency can effectively reduce fuel consumption and, thus, reduce agricultural carbon emissions [22]. ACESs can be effectively reduced by replacing chemical fertilizers with organic fertilizers and improving the fertilizer use efficiency through the upgrading of management technology [23].

Thirdly, research has focused on the factors influencing agricultural carbon emissions. Considering the fact that the application of chemical fertilizers and pesticides in agricultural production and the consumption of diesel oil by machinery are the leading factors increasing agricultural carbon emissions [24], the studies also found that economic and social factors such as the level of agricultural specialization [25], urbanization rate [26], agricultural technology [27], agricultural industrial structure [28,29], agricultural green production mode [30] and policy measures [18] can also affect regional agricultural carbon emissions. From the perspective of research methods, these studies can be roughly divided into the three categories described below.

The first type of research uses the factor decomposition method to investigate the factors driving agricultural carbon emissions. The factorization method includes the SDA model, Laspeyres statistical index, Kaya identity, LDA model, Shapley algorithm, IPAT model, STIRPAT model, etc. [31]. The factorization method can analyze the structure of complex technology–economy–social factors. In particular, the STIRPAT model can be expanded according to the agricultural production practices in the study area; thus, it is widely used [32]. However, the significance of the factors deconstructed through factor decomposition analysis to satisfy the identity relationship is weakened, and even their interpretation is one-sided. The second type of machine learning has inherent advantages in overcoming the disaster dimension through the mining of large amounts of data. For example, the random forest algorithm has the characteristics of robustness against noise and outliers and a good prediction stability; thus, it has been applied in research on factors influencing carbon emissions [33,34]. However, the random forest method causes the problem of overfitting due to the excessive complexity of the training stage, and there are certain requirements regarding the number of input variables [33]. The third type of research uses typical econometric analytical methods as research tools and general econometric methods including OLS, DID, etc., which offer the advantage of being able to explore the mechanisms and effects (positive or negative) of independent variables on agricultural carbon emissions [8,25]. However, the traditional measurement method assumes that the agricultural carbon effect is independent between regions, which is inconsistent with the specific agricultural production practice and weakens the practical significance of the conclusions.

As the undesirable output of agricultural production, agricultural carbon emissions are not simply a local environmental problem but spread to neighboring areas, to a large extent, through natural factors such as atmospheric circulation and atmospheric chemistry, as well as economic mechanisms, such as industrial transfer, industrial agglomeration and industrial structure. This requires local governments to adhere to the basic principle of combining territorial management with regional linkage in the process of agricultural carbon emission reduction and governance and to actively implement regional joint prevention and control policies and measures to investigate and control the inherent spatial correlation effect of agricultural carbon emissions. Undoubtedly, this can enhance the robustness of the analytical results and help to provide a more accurate decision-making basis. At present, a few studies have used spatial econometric analysis to carry out relevant empirical investigations [35,36].

The existing literature encompasses a great deal of research on agricultural carbon emissions, which provides a relevant research basis for this paper. However, most of the research on China’s ACESs and their influencing factors has been conducted on the national and provincial levels, with only a few regional cases, especially on the county level [9]. Compared with the county data, the errors in provincial or municipal data are larger, while township-level data are difficult to obtain. The county is a basic regional unit with a relatively independent form of administration and relatively complete regionalism and comprehensiveness. Each county has relatively consistent natural conditions and a consistent social, economic and cultural background, which often feature among China’s current statistical data. Therefore, it is more feasible to use the county as the vehicle of agricultural carbon emission reduction. Secondly, the above measurement methods for ACESs have their own advantages and disadvantages, their respective scopes for application and data requirements are quite different, and the results obtained using the different measurement methods are variable. In this paper, based on the availability of data and the characteristics of agricultural production in Zhejiang, a framework for estimating ACESs on the county level was established, and the method was used to calculate the ACESs on the county level in Zhejiang by combining the IPCC-recommended formula and the emission factor, accounting for the unique parameters of the Zhejiang region. Thirdly, this paper not only analyzes the temporal and spatial characteristics of agricultural carbon emissions but also uses the spatial Durbin model to analyze the spatial spillover effect of ACESs on the county level in Zhejiang Province and the mechanisms of the factors influencing agricultural carbon emissions. This marks a difference between this paper and the study of Hu et al. [32], focusing on the factors driving ACESs on the county level in Jiangsu Province.

Global climate change is a comprehensive problem at the intersection between nature and society. The carbon intensity is closely related to many aspects of social production and residents’ lives, and the key factors affecting carbon intensity vary across different stages of economic development. China is a vast country with immense differences in natural conditions, economic levels, agricultural structures and other aspects, and its agricultural carbon emissions have significant regional characteristics [9,12]. Zhejiang, as a developed coastal area in eastern China, has a congenital shortage of agricultural natural endowments. “Seven hills, one water and two fields” define the composition of the area, and the cultivated land area per capita is less than 1/4 of the national average level. The agricultural operation mode is diverse, the planting and breeding structure is complex, and hilly and mountainous counties account for the majority of the land. Zhejiang’s agricultural production has experienced a transition from high-input and high-carbon-emission to green and low-carbon agriculture. The development of green agriculture in Zhejiang is at the forefront of efforts in China. According to the results of the assessment of agricultural green development in provincial areas in 2016 by Wei Qi et al. [37], Zhejiang ranks first in China, with the highest comprehensive score. As the first experimental demonstration zone for the promotion of sustainable agricultural development in the whole province and the first pilot zone for the promotion of green agricultural development, Zhejiang plays a pivotal role in the process of sustainable agricultural development in the country. The analysis of Zhejiang Province can not only help us to explore the potential for agricultural carbon emission reduction and clarify the current characteristics of, and factors influencing, agricultural carbon emissions, as well as the degree of influence, but also has important practical and theoretical significance for the scientific development of regional agricultural carbon emission reduction measures. At the same time, it also helps to provide a tool for comparison and a reference for other provinces that can be used to realize a win-win situation combining agricultural development and carbon emission reduction.

For this reason, this paper takes 52 counties in Zhejiang Province, the first site of pilot areas for green agricultural development in China, as its research object and systematically investigates the measurement, influencing factors and spatial effects of ACESs in the unified system framework. First, from the perspectives of the four major aspects of rice planting, land management, livestock gastrointestinal fermentation and fecal emissions, a total of 20 greenhouse gas emission sources were constructed, the county agricultural greenhouse gas emission measurement framework was built, and the agricultural greenhouse gas emissions of the 52 counties (cities) in Zhejiang Province from 2014 to 2019 were calculated. Secondly, taking the intensity and total amount of carbon emissions as indicators, we analyzed the county-level gap and dynamic evolution characteristics of ACESs in Zhejiang Province, as well as those in the main food production areas, main food marketing areas and balanced production and marketing areas. The spatial-temporal characteristics of the agricultural greenhouse gas emissions in Zhejiang Province from 2014 to 2019 were analyzed using spatial statistics. Finally, based on the panel data of Zhejiang County, the key factors affecting the increase and decrease in ACESs in Zhejiang were analyzed.

## 2. Materials and Methods

### 2.1. Analytical Framework

There are several steps involved in the estimation of the spatial and temporal distribution characteristics of, and factors influencing, the carbon emissions from agricultural production in Zhejiang Province (Figure 1).

### 2.2. Research Methods

#### 2.2.1. Method for Estimating ACE Emissions

By referring to the results of research in the natural sciences, combining the results of several scholars [38,39,40,41,42,43,44] and considering the availability of county data, we established a county-level agricultural carbon emission measurement framework (see Table 1) based on four carbon sources: land management, livestock intestinal fermentation, manure management and rice planting. The second source refers to the carbon emissions caused by land management. Carbon emissions from the management of chemical fertilizers, pesticides, agricultural films, machinery and the ploughing of farmland were investigated. Thirdly, the intestinal fermentation of certain livestock and poultry in the process of livestock breeding can lead to methane emissions. Fourthly, the excrement of livestock and poultry causes methane and nitrous oxide emissions. Combined with the reality of Zhejiang, this paper focuses on cattle (including beef cattle, cows and buffalo), pig, sheep (including goats and sheep), rabbit and other major livestock and poultry breeds. The emission coefficient of each carbon source is the general coefficient and takes into account the local characteristics of Zhejiang. The calculation formula for the ACESs is as follows:(1)$$P=\sum_{1}^{i}C_{i}R_{i}GWP_{i}$$
where *P* is the total greenhouse gas emissions, *C_i_* is the *i*th source of greenhouse gas emissions, *R_i_* is the emission coefficient of the *i*th source of greenhouse gas emissions and *GWP_i_* is the contribution coefficient of the *i*th greenhouse gas. See Table 1 for the specific reference sources of the emission coefficients for each carbon source. It is worth noting that due to the differences in the water and heat conditions of different regions, the transplantation time and growth cycle of rice in different regions are variable, and different categories of early rice, middle rice and late rice are derived so that the corresponding methane emission levels of the paddy fields are also different. Therefore, the carbon emission coefficient of rice with the specific characteristics of Zhejiang Province elected for this paper to render the measurement results more accurate. Specifically, the carbon emissions caused by rice measured by referring to the carbon emission coefficient of Zhejiang derived from Wang Zhen et al. [25]. In order to facilitate the aggregation calculation, the 100-year global warming potential (*GWP*) of carbon dioxide is taken as the benchmark, with reference to its conversion coefficient, and various greenhouse gases are uniformly converted into the standard carbon equivalent. The carbon dioxide conversion coefficients of methane (CH_4_) and nitrous oxide (N_2_O) are 25 and 298, respectively.

As an indicator that can be used to measure the level of agricultural greenhouse gas emissions, the intensity of the agricultural greenhouse gas emissions is obtained using the ratio of regional agricultural greenhouse gas emissions to the gross agricultural product of the region. That is, *C_It_ = P_t_/S_t_* is the intensity of the agricultural greenhouse gas emissions in period *t*, *P_t_* is the agricultural greenhouse gas emissions in period *t* and *S_t_* is the gross agricultural product in period *t*.

The greenhouse gas emission structure refers to the proportion of the emissions from various sources among the total agricultural greenhouse gas emissions. That is, *R_i_ = P_i_/P*, where *P_i_* is the emissions of the *i* sources and *P* is the total agricultural greenhouse gas emissions.

#### 2.2.2. Method for the Spatial Autocorrelation Test Model

The Moran index is a commonly used index for exploratory spatial data analysis. Spatial correlation includes global correlation and local correlation, while the Moran index is commonly used in academia to analyze inter-regional correlation, in which the local Moran index [45] is used to study the correlation of each factor. Thus, in order to analyze the characteristics of the correlation between each county, this paper adopts the local Moran *I* index for the analysis. The Moreland index can reflect the spatial agglomeration characteristics of the variables by calculating the numerical variables.

The significant spatial mobility of agricultural production factors and the obvious inter-regional mobility of the agricultural output of China lead to the significant spatial correlation effect of agricultural production between regions. Whether or not the regional ACESs are spatially correlated and heterogeneous can be tested by describing the global Moran’s *I* index, which is an index of global spatial autocorrelation. Its expression is:(2)$$I=\frac{\sum_{i=1}^{n}\sum_{j=1}^{n}W_{ij}\left(x_{i}-\bar{x}\right)\left(x_{j}-\bar{x}\right)}{S^{2}\sum_{i=1}^{n}\sum_{j=1}^{n}W_{ij}}$$
where *x_i_* is the agricultural greenhouse gas emission intensity in region *i*; $\bar{x}$ is the sample mean value; *S*^2^ is the sample variance; and *W_ij_* is the element in row *i* and column *j* of the spatial matrix. The closer Moran’s *I* is to 1, the stronger the positive spatial correlation is. The closer Moran’s *I* is to −1, the stronger the spatial negative correlation is. Meanwhile, 0 means that there is no spatial autocorrelation.

It should be pointed out that the method of testing for spatial autocorrelation with the global Moreland index has great limitations. If there is a positive correlation (diffusion effect) between the ACESs in one part of the region and a negative correlation (agglomeration effect) between the ACESs in another part of the region, the residual value of the global Moreland index after the cancellation of the two may show that there is no spatial correlation between the regional agricultural carbon emissions, resulting in the failure of the test to scientifically reveal the local spatial correlation and heterogeneity in each direction. Therefore, it is necessary to further consider whether there is local spatial agglomeration of the observed values, which region contributes more to the global spatial autocorrelation and the extent to which the global evaluation of the spatial autocorrelation conceals local instability. We must also consider the possibility that the diffusion and agglomeration effects of the ACESs are not necessarily limited to the adjacent regions with common boundaries or regions situated over different distances. The local Moran’s *I* (spatial autocorrelation), local indicators of spatial association (LISA) and Moran scatter plot (Moran scatter plot) should be used to reflect the spatial autocorrelation (MSP) so as to more intuitively describe the local spatial interdependence and spatial heterogeneity characteristics of the agricultural output in different regions [46]. The calculation model of the local Moran’s *I* index is:(3)$$I=\frac{(X_{i}-\bar{X})}{\sum_{i=1}^{n}{\left(X_{i}-\bar{X}\right)}^{2}}\cdot \sum_{j\neq i}^{n}W_{ij}(X_{j}-\bar{Y})$$

In Formula (3), *W_ij_* is the normalized spatial weight matrix (the sum of each row is l). The expected value of the local Moreland index is:(4)$$E_{i}(I_{i}(=-\sum_{j=i}^{n}W_{ij}(n-1)$$

When the value of the local Moreland index *I_i_* is greater than its expected value *E_i_(I_i_),* this indicates that there is a spatial agglomeration phenomenon similar to the ACESs around region *i.* That is, there is a local spatial positive correlation tendency. When the value of the local Moreland index *I_i_* is less than its expected value *E_i_(I_i_),* this indicates that the ACESs of region *i* and its surrounding regions are significantly different. That is, there is a tendency towards local spatial negative correlation.

In addition, the Moran scatter plot (MSP) is a two-dimensional visualization of the *Z* and *ZW* data, where *Z* is a vector representing the deviation between the observed value and the mean value, and *ZW* is its spatially weighted average value, also known as the spatial lag vector. The formula for calculating the global Moran exponent in vector form is as follows:(5)$$I=\frac{n}{S_{0}}\frac{Z^{\prime }{W^{\prime }}_{Z}}{Z^{\prime }Z^{\prime }}$$

In Equation (5), $S_{0}=\sum_{i=1}^{n}\sum_{j=1}^{n}{W^{\prime }}_{ij}$. When ${W^{\prime }}_{ij}$ is the row-normalized space weight matrix, *S_0_ = n*, then the global Moreland index is the linear regression slope from *WZ* to *Z*. Standardizing the data (*Z*, *WZ*) in the Moreland scatter plot renders the Moreland index results comparable across different years. The regional units that have a strong influence on the global Moreland index can be diagnosed by standard regression. The first and third quadrants of the Moran scatter plot represent positive spatial correlation, while the second and fourth quadrants represent negative spatial correlation. The first quadrant represents the area unit with high observation value surrounded by the area of high value (high-high, high-high). The second quadrant represents the area unit of low observation value surrounded by the area of high value (low-high, low-high). The third quadrant represents the area unit of low observation value surrounded by the low-value area (low-low, low-low). Quadrant 4 represents the regions with high observed values surrounded by regions with low values (high-low, high-low). High-high and low-low clusters represent a typical positive local spatial correlation, while high-low and low-high outliers represent negative local spatial autocorrelation.

By combining the Moran scatter plot (MSP) with LISA agglomeration analysis, the local spatial autocorrelation and heterogeneity of the agricultural output in each region can be obtained.

#### 2.2.3. Spatial Dubin Model

As an external factor affecting economic development, carbon emissions, or air pollution, not only spread between regions with changes in natural climate conditions but also spread spatially through factor flow and industrial transfer, together with the development of intercity transportation infrastructure and communication technology. Therefore, carbon emissions may have a relatively obvious correlation effect on space. In addition, inter-regional growth competition indirectly leads to spatial correlation between inter-regional carbon emissions. On the one hand, under the pressures of growing competition and political promotion, the practice of lowering environmental standards and energy utilization intensity standards in one region to attract corporate investment and gain growth advantages may induce similar behaviors among local governments in other regions. On the other hand, efforts to reduce carbon emissions by strengthening environmental regulations in one region may lead to “free rider” behavior in the environmental governance of neighboring regions, which will lead to the increase in carbon emission levels in these neighboring regions. Any econometric test that ignores spatial correlation will fail to obtain a consistent estimation of the parameters. Therefore, we built a spatial econometric model to analyze the spatial correlation effects of ACESs and their influencing factors on the county level. The spatial Durbin model can effectively reflect the externalities and spatial spillovers caused by different influencing factors. The model includes the spatial correlation of both the dependent variables and independent variables. The model is as follows:(6)$$\ln I_{it}=a_{i}+\rho \sum_{j=1}^{i}W_{ij}WSQT_{it}+\beta X_{it}+\varphi \sum_{j=1}^{i}W_{jt}X_{jt}+\mu_{i}+\xi_{t}+\varepsilon_{it}$$
where *WSQTit* represents the equivalent agricultural greenhouse gas emissions of each county; *X* represents the factors affecting the agricultural greenhouse gas emissions (Table 2); *Wij* represents the spatial weight matrix; *β*, *ρ* and *φ* are regression coefficients; *μ_i_* and *ξ_t_,* respectively, represent regional and temporal fixed effects; *ε_it_* represents random disturbance items; *i*, *j* represents a geographical unit; and *t* represents the time variable.

With respect to the space-dependent spatial weight matrix selection, generally, spatial adjacency and spatial distance matrices can be used to achieve this [46]. In this paper, the traditional adjacent spatial weight matrix *(W1)* is used to construct the spatial weight matrix. That is, if two regions are geographically adjacent, the value is assigned as “1”. Otherwise, the value is referred to as “0”. At the same time, the inverse distance space weight matrix *(W2)* and inverse distance square weight matrix *(W3)* are selected as the alternative indexes of the robustness test, which is also convenient for overcoming the possible errors caused by the weight setting. These three weights are standardized and included in the spatial panel metrology model.

The factors influencing ACESs are very complex. They include natural conditions, agricultural management methods and social factors. In contrast to natural conditions and agricultural management methods, social factors affect agricultural greenhouse gas emissions on a macro-level. Based on the existing literature regarding the analysis of the factors influencing ACEs [24,25,26,27,28,29,32] and the availability of data, this paper selects the influencing factors from the technical, economic and social perspectives, in which the technical factors are expressed in the total power of the agricultural machinery (AM), and the economic factors include the developmental level of the agricultural economy (AGDP), structure of the agricultural industry (Str), crop planting structure (Cps) and agglomeration degree of the agricultural industry (IA). The social factors include the urbanization rate (Urban), rural population (RP), disposable income of the rural residents per capita (INC), total power of the agricultural machinery (AM) and human capital (Edu).

(1)Agricultural technology factor (AM). The AM is the technical factor, representing the level of agricultural technological development. Advances in agricultural technology can improve the efficiency of the machinery used, which will then produce fewer carbon emissions on the same level as the output, expressed in terms of the total agricultural machinery power [47].(2)Agricultural industry structure (Str). The industrial structure of the agricultural sector has a direct relationship with carbon emissions. Compared with the forestry and fishery industries, plantation and livestock farming contribute the major share of carbon emissions [48](3)The agricultural development level (AGDP). The agricultural development level is the key factor affecting agricultural carbon emissions, and the agricultural economic development level is the main factor driving greenhouse gas emissions, but whether or not it will drive the growth in greenhouse gas emissions depends on the quality and stage of economic development [26]. Therefore, the agricultural development level is also taken as an important explanatory variable in this paper, and the gross agricultural output value per capita is taken as an indicator, specifically the ratio of the gross agricultural output value to the total rural population.(4)Crop planting structure (Cps). The input and carbon emissions of grain and cash crops differ substantially [49]. In view of the fact that rice plays a leading role in contributing to ACESs in Zhejiang, this paper uses the proportion of grain crop sowing land in the total agricultural planting area to characterize the crop planting structure.(5)Industrial agglomeration (IA). Industrial agglomeration generally refers to the proximity of related activities in the same industry in a specific geographical space [50]. Studies have found that industrial agglomeration can have either positive or negative impacts on industrial development, thereby validating the Williamson hypothesis [51]. The measurement of the level of industrial agglomeration has been widely employed in research on the regional economy, resources and environment, as well as low-carbon production. The measurement methods include the industry concentration, Herfindahl–Hirschman index, location quotient, etc. Considering the availability of data and the benefit of eliminating regional-scale differences, this paper uses the location quotient to measure the level of industrial agglomeration. At the same time, in order to examine whether there is a nonlinear relationship between the level of industrial agglomeration and agricultural carbon emissions, this paper adds a quadratic term of the level of industrial agglomeration as an explanatory variable. The specific expression of the location quotient is $IA_{ab}=\left(P_{ab}/P_{b}\right)/\left(P_{a}/P\right)$. Here, $IA_{ab}$ represents the location quotient of the agricultural industry in region *b*, $P_{ab}$ represents the number of primary industry employees in region *b*, $P_{b}$ represents the total number of primary, secondary and tertiary industry employees in region *b*, $P_{a}$ represents the number of primary industry employees in China and *P* represents the total number of primary, secondary and tertiary industry employees in China.(6)The rural population (RP). The rural population is an important factor influencing agricultural carbon emissions, and the research basically confirms that there is a positive relationship between the two. The more people employed in agriculture there are, the greater the agricultural carbon emissions will be, and the opposite is also true [52].(7)The urbanization rate (Urban) is a social concern factor. The urbanization rate has an uncertain effect on agricultural carbon emissions. On the one hand, with the increase in the urbanization rate, a large flow of rural labor moves into the cities, promoting large-scale and intensive agricultural production, improving labor productivity, resource utilization and green production efficiency, and reducing agricultural carbon emissions [26]. On the other hand, urbanization causes the agricultural labor force to acquire the characteristics of aging, feminization and part-time employment. In order to avoid reductions in agricultural production, farmers may increase their use of alternative labor factors such as chemical fertilizers, pesticides, agricultural film and mechanical facilities, thus increasing agricultural carbon emissions [53]. Therefore, we used the urbanization rate as a control variable, which is measured as the proportion of the urban population to the total population.(8)The disposable income of rural residents per capita (INC). The INC is a social concern factor. The impact of the INC on ACEs may follow the environmental Kuznets curve (EKC). The increase in the disposable income of rural residents per capita makes it possible for farmers to expand the input of the means of agricultural production such as pesticides and fertilizers and expand the scale of planting, thus producing an increase in greenhouse gas emissions. On the other hand, to a certain extent, the income level of farmers also reflects the ability of the farmers to pay for green production technology, and the increase in the farmers’ income may also encourage farmers to adopt green production technology for low-carbon production [23].(9)Human capital (Edu). The improvement of human capital can help agricultural producers to use environmentally friendly production technologies in order to implement the development of low-carbon agriculture [53]. Therefore, this paper analyzes whether human capital contributes to the reduction in agricultural carbon emissions. Due to the limited data available, we can assume that the number of students in ordinary middle schools is equal to the number of students per 10,000 people.

### 2.3. Study Area and Data Sources

The spatial units of this study are 52 counties (cities) in the county-level administrative divisions of Zhejiang Province (referred to as counties or regions). In 2019, Zhejiang Province had 11 prefecture-level cities, 20 county-level cities, 33 counties (one of which is an ethnic autonomous county) and 37 municipal districts. Considering that the economic development level and urbanization level of the municipal districts in Zhejiang Province are generally high, most have historically been managed as urban areas, and the proportion of agriculture is low. Thus, in this paper, the municipal districts are excluded. As Longgang City was not established until 2019, the preliminary data are lacking, and it is not included here. Therefore, the spatial units examined in this study are 52 counties (19 county-level cities and 33 counties). The study period lasted from 2014 to 2019, with a total of 312 samples. Figure 2 depicts the spatial distribution of 52 counties in Zhejiang Province.

The original agricultural carbon emission data and rural population data used in this paper were derived from the Agricultural Statistical Data of Zhejiang Province from 2014 to 2019. In contrast to the general public statistical yearbooks, the Agricultural Statistical Data of Zhejiang Province contains detailed information, including the production of early rice, mid-season rice and late rice. Detailed data on the use of nitrogen fertilizer, phosphate fertilizer, potash fertilizer and compound fertilizer, as well as animal husbandry, can help improve the accuracy of the measurement of agricultural carbon emissions. The Agricultural Statistical Data of Zhejiang Province refers to the internal data of the Department of Agriculture and Rural Affairs of Zhejiang Province. At present, only the paper version from 2014 to 2019 can be obtained and manually input and collated. We processed the data in advance by referring to the literature. The average annual quantity of livestock feed with an output rate of more than 1 adopted (output quantity of the year * average life cycle) was /365.

The data on the influencing factors, such as the agricultural development level (AGDP), urbanization rate (Urban) and disposable income of rural residents per capita, were obtained from the Zhejiang Statistical Yearbook, representing data collected over the years. In order to eliminate the interference of price factors, the total agricultural output value was converted by taking the 2014 price as the constant price. Table 2 reports the descriptive statistical analysis of each variable.

## 3. Results

### 3.1. Analysis of the ACEs

#### 3.1.1. Time Evolution Characteristics of the County’s Agricultural Carbon Emissions

According to Equation (1), the ACEs in Zhejiang Province from 2014 to 2019 were measured (Table 3, Figure 3). The results show the following:

(1)Total emissions. On the whole, the agricultural greenhouse gas emissions in Zhejiang Province showed a downward trend from 2014 to 2019 and only rebounded slightly in 2018. In 2019, the total agricultural greenhouse gas emissions of Zhejiang County were 8.5882 million tons, a decrease of 18.13% compared to 2014. In 2019, the agricultural greenhouse gas emissions caused by rice planting, gastrointestinal fermentation, fecal management and land management in Zhejiang Province were 5,160,100 tons, 304,400 tons, 876,900 tons and 2,246,800 tons, respectively, accounting for 60.08%, 3.54%, 10.21% and 26.16%.(2)Emission intensity. From 2014 to 2019, the agricultural carbon emission intensity in Zhejiang Province was generally on the rise. In 2019, the agricultural carbon emission intensity in Zhejiang Province was 1.303 tons/CNY 10,000, an increase of 4.864% compared to 2014 and 2.31 tons/CNY 10,000 lower than the national agricultural carbon emission intensity in the same period.(3)From the change in the agricultural carbon emission structure, we can observe that the proportional agricultural carbon emission structure from 2014 to 2019 was relatively stable, and rice planting was the largest emission source (accounting for 56~60%). The second significant source was carbon emissions caused by land management, accounting for 24~27%. The carbon emissions caused by gastrointestinal fermentation and fecal excretion of livestock and poultry accounted for 13–19%. Among these, rice planting was the largest carbon emission source among Zhejiang’s agriculture. In 2019, the carbon emissions from rice planting reached 5,160,100 tons, a decrease of 796,000 tons compared with 2014. The greenhouse gas emissions caused by land management showed a slight upward trend, and the proportion increased from 24.82% in 2014 to 26.16% in 2019, rendering it the second-largest agricultural carbon source after rice planting. The greenhouse gas emissions caused by the gastrointestinal fermentation of livestock in animal husbandry and livestock manure management showed a downward trend. This decrease in livestock husbandry emissions is mainly due to the decrease in the number of livestock raised. Taking live pigs as an example, the number of live pigs in Zhejiang Province decreased annually from 2014 to 2019 and had decreased by 55.7% in 2019 compared with 2014.

#### 3.1.2. Spatial Evolution Characteristics of County-Level Agricultural Carbon Emissions

The total amount and intensity of the agricultural greenhouse gas emissions from each county of Zhejiang Province are shown in Table 4. It can be seen from Table 4 that the agricultural greenhouse gas emission intensities of the counties varied greatly in 2019. Among them, Longyou County ranks first in terms of its agricultural greenhouse gas emission intensity, which is 4.965 tons/CNY 10,000. However, Pan’an County has the lowest agricultural greenhouse gas emission intensity, which is 0.330 tons/ CNY 10,000, being 14 times higher than that of Pinghu. In this paper, the 52 counties are divided into three echelons according to the intensity of the agricultural greenhouse gas emissions.

The first echelon is the area where the agricultural greenhouse gas emission intensity exceeds 1.5 tons/CNY 10,000, comprising 15 counties and cities including Longyou County, Deqing County, Pinghu City, Haiyan County, Pingyang County, Jiangshan City, Yongjia County, Changshan County, Tonging City, Qingyuan County, Haining City, Taishun County, Cangnan County, Yongkang City and Longquan City. This echelon is mainly in the southwest and northern areas, concentrated in Wenzhou, Jiaxing and Quzhou. The reason for this spatial pattern may be that the planting area of the food crops in these two regions accounts for a large proportion of the land, and the prices of the food crops are relatively low compared with those of cash crops. Thus, the overall carbon emission intensity is high.

The second echelon is the area where the agricultural carbon emission intensity is between 1 and 1.5 tons/CNY 10,000, comprising 22 counties and cities including Qingtian County, Lanxi City, Ninghai County, Yueqing City, Rui’an City, Jingning County, Kaihua County, Tiantai County, Suichang County, Jinyun County, Wuyi County, Xianju County, Yunhe County, Shengsi County, Dongyang City, Wencheng County, Xiangshan County, Pujiang County, Changxing County, Jiashan County, Zhuji City and Sanmen County. It is mainly located in the central area of Zhejiang, concentrated in Jinhua City and partially in Taizhou City and Lishui City.

The third echelon is the area where the agricultural carbon emission intensity is less than 1 ton/CNY 10,000, comprising 15 counties and cities including Daishan County, Linhai County, Anji County, Wenling City, Songyang County, Shengzhou City, Jiande City, Tonglu County, Yuyao City, Yuhuan City, Cixi City, Chun’an County, Yiwu City, Xinchang County and Pan’an County. This echelon is mainly in the central and western regions and the eastern coastal areas.

To cite a few examples, we can observe that Ningbo, Taizhou and other coastal areas have developed vegetable and fruit planting, so that the input of the agricultural film and fertilizer consumption is large, thus increasing the agricultural carbon emissions. On the other hand, the high degree of mechanization leads to a high consumption of diesel and other energy, which also promotes the growth in agricultural carbon emissions. The low-emission areas are mainly concentrated in southwest Zhejiang (Quzhou, Lishui) and Huzhou. Compared with the coastal areas, these areas use less pesticides, fertilizers and diesel and are rich in forest and farmland resources, which can effectively play a role in carbon sequestration. Therefore, their ACESs are low.

### 3.2. Spatial Correlation Analysis

#### 3.2.1. Global Moreland Index Analysis of Spatial Correlation of Agricultural Carbon Emissions

Table 5 reports the results of the global spatial autocorrelation analysis calculated using three spatial weight matrices: the adjacency matrix (*W1*), inverse distance weight matrix (*W2*) and reciprocal geographical distance square matrix (*W3*). Assuming that there is no spatial autocorrelation between the ACESs among the counties, the global Moreland index is calculated based on the three spatial weight matrices. According to the null hypothesis, p is less than or equal to 5%, and the absolute value of z is greater than 1.96. Thus, the estimated results show that the larger the value is, the more obvious the spatial correlation will be. Therefore, the Moreland index calculated based on the adjacency weight matrix has the largest and most significant spatial autocorrelation. It passes the test with a significance level of 5%, indicating that the spatial effect of agricultural carbon emissions on the county level is highly apparent. Therefore, the spatial effect of the agricultural greenhouse gas emissions in Zhejiang Province cannot be ignored.

#### 3.2.2. Local Moreland Index and LISA Analysis of the Spatial Correlation of Agricultural Carbon Emissions

The test results of the global Moreland index in Table 1 show that, on the whole, the ACESs of each county and city show significant spatial autocorrelation, but the global Moreland index cannot show the local spatial agglomeration and local spatial autocorrelation characteristics of the ACESs on the county level. Therefore, in this paper, the Moreland scatter plot (MSP) was used to further reveal the local spatial characteristics of the ACESs on the county level. The results are shown in Figure 4.

Table 6 shows the regional distribution of Moran’s *I* during 2014–2019, and the results show that the northeast of Zhejiang Province is mainly in the first quadrant HH region, while the southwest is mainly in the LL region, showing a significant positive correlation. Changshan, Deqing, Pujiang, Suichang, Tonglu, Yiwu and other counties and cities are in the LH region, and Cangnan, Jiangshan, Linhai, Ninghai, Zhuji and other places are in the HL region. The spatial heterogeneity of the GHG agricultural emissions in Zhejiang Province is evident. In addition, most of the counties and cities in Zhejiang are still in high-emission areas and have not changed for a long time. Therefore, reducing the agricultural greenhouse gas emissions of the counties of Zhejiang Province is an important proposition for the sustainable development of Zhejiang’s agriculture.

### 3.3. Spatial Econometric Estimation and Results of the Analysis of the ACESs and Their Driving Factors

#### 3.3.1. Estimation Results of the Spatial Dubin Model

From the previous section, it can be seen that there is a certain spatial correlation between the agricultural greenhouse gas emissions. By establishing a spatial measurement model, we can measure the specific direction and extent of the impacts of the major factors on agricultural carbon emissions more accurately. Before the model estimation, we used the natural logarithms of the GDP and income to stabilize the data. As the research data are short panel data, neither a unit root test nor a cointegration test is required. The maximum value of the variance expansion factor of the independent variable is 8.320, which can evade the multicollinearity problem.

Before the analysis of the factors influencing agricultural carbon emissions, a global spatial autocorrelation test of the ACESs and various influencing factors was conducted using Stata17 software(StataCorp, Lakeway Drive, TX, USA) and the global Moran index. The results are shown in Table 7. As can be seen from Table 7, the Moreland indexes of the ACESs in the counties during 2014–2019 all passed the significance test at the *p* < 1% level. Among the factors influencing agricultural carbon emissions, in most years, the Moran index also passed the significance test, indicating that the ACESs are driven by different factors and that there is the possibility of cross-county spatial spillover effects. Therefore, it is necessary to further explore the factors influencing ACESs and their potential indirect impacts.

In this paper, the Lagrangian multiplier (LM) test, likelihood ratio (LR) test, Hausman test and Wald test are applied to Equation (3) in order to determine the specific estimation form of the spatial econometric model. First, the LM test is used to judge the type of spatial effect. Using the LM (error) and LM (lag) to test the spatial lag and spatial error models, the results show that the spatial lag model is better when it passes the chi-square test at the 1% significance level. Secondly, with respect to the model comparison, the test results of the LR (sdm sar) and LR (sdm sem) are also significant at the level of 1%, indicating that the spatial Dubin model is superior to the spatial lag model and the spatial error model. Thus, the spatial Dubin model is more suitable for this paper. At the same time, the Hausman test was used to determine the selection of either the fixed effect or random effect model. The results showed that the fixed effect model was significantly better than the random effect model at the level of 1%. Therefore, this paper uses the spatial Dubin fixed effect model to perform the estimations.

According to the aforementioned Equation (3), spatial Durbin model regression was conducted using Stata 17.0 to analyze the factors influencing the county ACESs in most regions of Zhejiang Province. The individual and time double fixed effects were uniformly selected, and the results are shown in Table 6 below.

Table 8 shows the spatial spillover coefficients under the three spatial weight matrices *W(1)*, *W(2)* and *W(3)*. The estimated values of are all positive at the significance level of 1%, which indicates that in the sample area, spatial spillover is an important factor in promoting the growth in agricultural carbon emissions. At the same time, the significance and coefficient direction of each variable under the three spatial weight matrices are basically consistent, indicating that the regression results are relatively robust.

#### 3.3.2. Direct Effect and Spillover Effect Analysis

Elhorst [54] pointed out that when the global effect is included in the setting of the model, the point estimation results of the spatial econometric model itself do not represent the marginal impacts of the explanatory variables. Therefore, to compare and analyze the difference in the effect of each explanatory variable and its spatial spillover effect, it is necessary to further measure the direct and indirect effects of each explanatory variable based on the point estimation results of the model. LeSage and Pace [55] also pointed out that analyzing the influences of explanatory variables on the explained variables and their spatial spillover effects based solely on the point estimation results of the spatial Doberman model or the dynamic spatial Doberman model itself may lead to incorrect conclusions. In order to accurately determine the spatial spillover effect of each variable on the agricultural carbon emissions, this paper refers to LeSage and Pace [55]. According to the parameter estimation results displayed in Table 6, the direct effects, indirect effects and total effects of the explanatory variables in the spatial model were further estimated (see Table 9). Among these, the direct effect refers to the influence of the independent variable on the dependent variable in the region, including the direct influence yielded by the variable and the influence fed back into the region through the impact on the adjacent region (feedback effect, feedback effects, FE). Indirect effects, also known as spatial spillover effects, represent the impact of an explanatory variable in the “adjacent” region on the explanatory variables in the region under analysis. The total effect is the sum of the direct effect and indirect effect, which can be interpreted as the average impact of the change in an explanatory variable in one region on the explanatory variables in all the regions.

As can be seen from Table 7, the direct and indirect effects of the level of agricultural economic development (lnAGDP), agricultural technology factor (AM) and the level of urbanization (Urban) on the ACESs failed to pass the significance test. The possible reason for this result is that the levels of agricultural development and urbanization among the counties in Zhejiang Province are not balanced. Rather, the difference is large, and the average effect is not significant. The crop planting structure (Cps) is not conducive to the reduction in local agricultural carbon emissions and has no significant spatial effect on the neighboring counties. The level of rural income per capita (lnINC) has a significant inhibitory effect on the ACESs in this county and its neighboring counties. With the increase in disposable income per capita, people attach more importance to the ecological environment. At the same time, the increase in income is conducive to farmers’ adoption of low-carbon agricultural production technologies and the reduction in agricultural carbon emissions. The primary coefficient of the direct effect of industrial agglomeration (IA) is negative, the secondary coefficient is positive and significant, and the spatial spillover effects of the primary and secondary coefficients are not significant, which indicates that the agricultural industrial agglomeration in the adjacent areas does not have significant spatial spillover effects on the local agricultural carbon emissions. This shows that there is a U-shaped relationship between the level of industrial agglomeration and agricultural carbon emissions. That is, with the improvement in the level of agricultural industrial agglomeration, the ACESs underwent a process of first reducing and then increasing. On the one hand, industrial agglomeration has obvious Marshall externalities, which can achieve large-scale effects, strengthen the spillover of knowledge, help to achieve agricultural scientific and technological progress and, thus, effectively reduce carbon emissions [56]. However, excessive industrial agglomeration results in a crowding effect, which increases the demand and consumption of production factors, thus increasing carbon emissions [57]. The spatial spillover effect of the human capital level (lnEdu) on the ACESs is not significant, but the direct effect is significant. The spatial spillover effect of human capital level (lnEdu) on ACES is not significant, but the direct effect is significant. The possible explanation is that a county with a large number of ordinary middle school students is likely to be a county with a high proportion of agriculture. A county with a high proportion of agriculture tends to show a high level of agricultural carbon emissions; thus, its direct effect is significant. The human capital represented by ordinary middle school students has little attraction to highly skilled talents in neighboring areas. The improvement of human capital level in neighboring counties has not promoted the reduction of agricultural carbon emissions in neighboring counties through the inter-regional “demonstration imitation” mechanism; thus, its spatial effect is not significant. Due to the limited availability of data, there are some limitations in taking the number of students in ordinary middle schools as a variable to measure the level of human capital.

#### 3.3.3. Analysis of the Regional Regression Results

Previous studies have shown that the level of economic development is closely related to agricultural carbon emissions. In order to further deepen our understanding of the impact of economic development on agricultural carbon emissions, we conducted grouping analysis based on the differences in the economic development level. In practice, Zhejiang Province is usually divided into 26 relatively underdeveloped counties (economically backward counties, hereinafter referred to as the “26 mountainous counties”) and non-mountainous counties according to the differences in the level of economic development between the counties. Since 2015, Zhejiang has implemented a differentiated assessment system for these two counties with different economic development orientations. It has abandoned the orientation of “GDP first” and “economic development as the focus” for the 26 mountainous counties, cancelled the GDP assessment, and implemented the performance assessment method of prioritizing agriculture and ecological protection. The degree of economic intervention affects the interaction and value orientation of the competition between local governments. In order to verify the heterogeneous impact of the economic development level and the heterogeneous impact of the performance appraisal methods, we further regressed the 26 mountainous counties and non-mountainous counties, respectively. The measurement results of the spatial Dubin model are shown in Table 10, and the deconstruction of the grouping effect is shown in Table 11 and Table 12. With respect to the spatial spill coefficients of the two sample groups ρ, the estimated values of are positive at the significance level of 1%, which indicates the spatial spillover of the ACESs in the two sub-sample regions. In the following paragraphs, we use the breakdown effect results (Table 11 and Table 12) to analyze the differences in the effects of the explanatory variables and their spatial spillover effects.

In the case of the full-sample estimation results, the coefficients of the direct effect, indirect effect and total effect of the agricultural economic development level (lnAGDP) are positive, but they are not significant. However, in the grouping regression, in the sample group of the 26 counties in the mountainous area, the direct effect parameter of the agricultural economic development level is estimated as not significant, while the indirect effect is significantly positive at the level of 5%, indicating that, although the agricultural economic development level has no significant impact on the ACESs in the region, it has a significant spatial spillover effect on the adjacent areas. In the sample group of the 26 non-mountainous counties, the direct effect parameter of the agricultural economic development level (lnAGDP) is estimated to be significantly positive at the level of 5%, and the indirect effect is significantly positive at the level of 1%, indicating that within the non-mountainous counties, the growth of the agricultural economy not only promotes the increase in ACESs in the region but also has a significant spatial spillover effect on the adjacent areas, leading to the increase in ACESs in the adjacent areas. That is, the local efforts to promote agricultural economic growth have a “demonstration effect” on the neighboring regions. The strategic behavior of local governments in the context of growing competition causes a region to regard its neighboring regions with rapid agricultural economic growth as “models”. In order to pursue the growth of the agricultural economy, it is possible to select the extensive agricultural growth mode driven by factor inputs that can achieve rapid results in the short term, chiefly by increasing the use of fertilizer. The increase in the agricultural output is obtained by increasing the input of pesticides, which is accompanied by the increase in agricultural carbon emissions. In terms of the overall effect, the agricultural economic development level of the two groups of samples significantly promotes the increase in agricultural carbon emissions. This conclusion, combined with the descriptive analysis of the annual decline in the total amount of ACESs and the fluctuating decline in their intensity in Zhejiang, means that, as a whole, Zhejiang’s county-level agricultural economic growth mode has not yet turned to the green, low-carbon growth path but is in a period of transition from extensive agriculture to intensive agriculture.

The impact of the rural disposable income per capita (lnINC) on the ACESs showed heterogeneity in the two sub-sample groups. In the 26 mountainous counties, the coefficient of the direct effect of the rural disposable income per capita on the ACESs is positive, while the coefficient of the indirect effect is negative, but both are insignificant. In the sample group of the non-mountainous counties, the direct effect parameter of rural disposable income per capita is estimated to be significantly negative at the level of 1%, while the indirect effect is significantly positive at the level of 1%. This indicates that, although the variable of rural disposable income per capita can inhibit the ACESs in this region to a certain extent, it has a significant positive spatial spillover effect on the adjacent areas, leading to the increase in the ACESs in those adjacent areas. The direct effect and indirect effect of the two sample groups are in the opposite directions, indicating that the rural disposable income per capita can only inhibit the ACESs after reaching a certain level. With the increase in the disposable income per capita, people pay greater attention to the ecological environment, and their ability to pay for relatively advanced low-carbon agricultural technologies and low-carbon agricultural production materials will be enhanced.

Urbanization (Urban). In the sample group of 26 counties in the mountainous area, the direct benefit coefficient is negative but not significant, and the indirect effect coefficient is significantly negative, indicating that urbanization has not significantly improved the low-carbon development level of agriculture in the region through the transfer of the surplus agricultural population. Instead, it has had the same spillover effect on the adjacent areas. That is, the increase in the level of urbanization in the adjacent areas has had an inhibitory effect on the ACESs in the region. The increase in the level of urbanization in the adjacent areas can attract the transfer of surplus local agricultural labor, thus improving local agricultural production efficiency and reducing local agricultural carbon emissions. In the sample group of non-mountainous counties, the impact of urbanization on the ACESs in these counties and the surrounding counties is not significant.

Planting structure (Cps). In the two sample groups, the coefficient of the direct effect of the crop planting structure is significantly positive; that is, the greater the proportion of the grain planting area in the total crop planting area is, the greater the agricultural carbon emissions will be. The reason for this trend is that grain is a more important source of ACESs than other crops, especially rice planting. From the above analysis, it can be seen that rice planting is the largest source of agricultural carbon emissions in Zhejiang. The spatial benefits of the planting structure in the two sample groups show heterogeneity; that is, the spatial effect of the planting structure on the ACESs is significant in the 26 mountainous counties but not significant in the non-mountainous counties. The possible reason for this heterogeneity is that Zhejiang has adopted a differentiated assessment system for the two sample regions. In 2015, Zhejiang abandoned the orientation of “focusing on GDP first” and “focusing on economic development” for the 26 mountainous counties in order to accelerate development, cancelled the GDP assessment, and implemented the performance assessment method of prioritizing agriculture and ecological protection. In the 26 counties in the mountain area, the planting structure has a significant spatial effect on the agricultural carbon emissions; that is, the increase in the grain planting area in the adjacent areas will lead to the increase in ACESs in this region. A possible explanation for this pattern is that in the context of China’s food security strategy, the adjustment and change in the grain sown area is not entirely purely a market behavior. The Party Central Committee requires that “the Party and the government should share the responsibility for food security”. In this situation, the 26 counties in the mountainous areas, as regions with relatively backward economies and relatively high proportions of agricultural land, will be rewarded or subsidized by the superior government for the increase in the grain sown area in the adjacent areas. This will have a demonstration effect on the local area and increase the proportion of the local grain sown area, thus leading to the increase in agricultural carbon emissions. However, in the non-mountainous counties, the spatial effect is not significant. A possible explanation for this trend is that the non-mountainous counties still follow the macro-institutional initiative of “focusing on economic construction”, mainly the incentive mechanism for the promotion of local officials to tournaments based on GDP assessment [58]. In addition, the contribution of agriculture to GDP is very small. In non-mountainous counties, the benefits of administrative assessment yielded by improving the grain planting area are not enough to form a model. Rational neighboring areas will not increase the grain planting area as much as is possible and will pay more attention to other production activities that contribute more significantly to GDP. This also provides an explanation for the fact that the “de-grain” phenomenon is more likely to occur in more economically developed regions.

The direct effects of the primary and secondary terms of the industrial agglomeration level (IA) on the two sample groups are not significant. However, the coefficient symbols of the two groups are opposite to one another. The spatial spillover effect of industrial agglomeration is not significant. The possible explanation for this is that the alternative indicator of agricultural industry agglomeration, in this paper, is the location quotient expressed by the ratio of the total number of employees of the primary, secondary and tertiary industries in Zhejiang to that of the whole country. Zhejiang has a developed non-agricultural economy. With the restriction of the natural endowment of mountainous agriculture, the agricultural industry agglomeration in Zhejiang is relatively low and tends to weaken. Thus, the direct effect and indirect effect on ACESs are not significant.

The lnRP of the two sample groups is heterogeneous. In the sample group of the 26 counties in the mountainous area, the estimation of the direct effect parameters of the lnRP did not pass the significance test, but the indirect effect was significantly negative. This indicates that although the rural population did not have a significant impact on the reduction in the agricultural carbon emissions of the county, it had a significant effect on carbon emission reduction in the neighboring counties. The overall effect was significantly negative, and the rural population generally contributed to the reduction in the carbon emission levels of the 26 counties in the mountain area.

In the sample group of non-mountainous counties, the direct effect of the lnRP is significantly positive, but the indirect effect is not significant, indicating that the rural population has a promoting effect on the ACEs in the county and has no significant impact on the neighboring counties. This may be explained by the fact that most of the 26 counties in the mountainous area are agricultural counties, and there are a large number of agricultural laborers among the rural population, some of whom travel to the neighboring counties, promoting the reduction in agricultural carbon emissions. In the economically developed non-mountainous counties, the secondary and tertiary industries are developed, and the rural population has less of its labor force engaged in agricultural production. This labor force may be replaced by machinery, thus promoting the increase in local agricultural carbon emissions.

## 4. Discussion

This paper studied the ACESs of Zhejiang Province on the county spatial scale, constructed a county-level ACES inventory, calculated the ACESs of 52 counties in Zhejiang Province, and explored the driving factors and spatial effects influencing the county’s agricultural carbon emissions. Compared with the existing research, which has mainly focused on the national and provincial scales, this paper estimated the ACESs in Zhejiang Province based on county data and analyzed the spatial effects driving the evolution of the ACESs in the county from the perspectives of the economic development level and industrial agglomeration. Compared with the results regarding national carbon emissions from the research of Liu M. and Yang L. [9], this paper used more detailed calculation data of the ACESs in the county, and the spatial concentration of ACESs in the county showed a stronger distribution trend. In addition, compared with Liu, M. and L. Yang [9], Cui, Y. et al. [12] research on the spatial spillover effect of agricultural carbon emissions, this paper further discussed the heterogeneity of the spatial spillover effects and influencing factors of ACESs on the county level, providing a theoretical basis for Zhejiang Province that can be used to formulate agricultural carbon emission reduction policies tailored to local conditions according to differences in economic development levels. However, there are still some limitations to this study. Firstly, the mainstream emission parameter method is mainly used for agricultural carbon emission measurements. Most of the parameters are based on the emission factors recommended by IPCC and FAO. Only rice is based on independent local emission factors. Nevertheless, the emission factors of the IPCC method are too generic, lacking the necessary accuracy and national specificity. Secondly, in terms of the carbon sources, this paper accounted for four major carbon sources: rice planting, land management, the gastrointestinal fermentation of livestock and fecal emissions (livestock and poultry breeding). Due to the lack of data on the county level, this paper could not investigate the dynamic change in carbon emissions caused by straw returning over the years, which could represent a direction for further research in the future.

Combined with the limitations of the current research, in order to further promote research on agricultural carbon emissions, the theoretical mechanism of ACES reduction should be systematically and scientifically constructed. Prospects for future research relate to the following aspects: the summarization and integration of the existing agricultural greenhouse gas emission measurement methods, the determination of the appropriate small-scale parameters of agricultural carbon emissions, the preparation of county-level and smaller-scale agricultural carbon emission measurement systems, and the development of methods to avoid the shortcomings of the various existing methods in terms of their accuracy and breadth. Economic thought is deeply embedded in the research on agricultural carbon emissions, and the reduction in agricultural carbon emissions and agricultural economic development are coordinated by economic means. Efforts should be undertaken to provide data support and decision-making support to the agricultural sector in order to mitigate and adapt to climate change.

## 5. Conclusions

Based on the panel data of the Zhejiang counties in China from 2014 to 2019, we measured the ACESs of the Zhejiang counties using the emission coefficient method. Adopting exploratory spatial statistics and spatial measurement methods, we studied the temporal evolution characteristics of, and factors influencing, the ACESs of the Zhejiang counties and discussed the spatial spillover effect. The main research conclusions are as follows:

Firstly, from 2014 to 2019, the ACESs in Zhejiang Province showed a downward trend on the whole. Among the four major carbon sources, rice planting was the largest carbon source of Zhejiang’s agricultural carbon emissions, accounting for an average of 59.07%. The average carbon emissions caused by land management, fecal management and gastrointestinal fermentation accounted for 26.17%, 11.35% and 3.56% of the ACESs, respectively. The level of intensity of the ACESs fluctuated on the whole, showing an obvious agglomeration effect.

Secondly, the increase in ACESs in this county promotes the increase in ACESs in the neighboring counties, with a positive spatial spillover effect. The estimation results of both the full samples and grouped samples confirm this conclusion.

Thirdly, the spatial effect mechanism of the county-level ACESs is heterogeneous. The level of agricultural economic development is the main factor driving the growth in agricultural carbon emissions. The cluster estimation results of the impacts of rural disposable income per capita, urbanization, the planting structure and other factors on the ACESs are heterogeneous.

## 6. Suggestions

Countermeasures and suggestions. (1) We should build a regional coordinated governance mechanism for ACESs on the county level. We should try to establish a regional carbon compensation and carbon trading system according to the level of economic development, promote the coordinated development of a regional green economy, and, at the same time, actively create favorable conditions in order to fully highlight the county’s own experience so as to drive the reduction in carbon emissions caused by agriculture in the neighboring counties. (2) We should optimize the agricultural industrial structure. This study found that the carbon emissions caused by rice planting accounted for more than 50% of the ACESs in the province. Thus, we must pay attention to the reduction in carbon emissions from rice planting. The empirical results show that reducing the proportion of the grain sown area in the total sown area has positive significance as a method for reducing agricultural carbon emissions. However, it should be noted that the internal industrial structure of agriculture can only be properly adjusted on the premise of ensuring food security. (3) To formulate the agricultural carbon emission reduction strategy based on local conditions, the carbon emission reduction strategy should be formulated according to the county’s individual stage of economic development and the actual state of its agricultural development.

## Figures and Tables

**Figure 1 ijerph-20-00189-f001:**
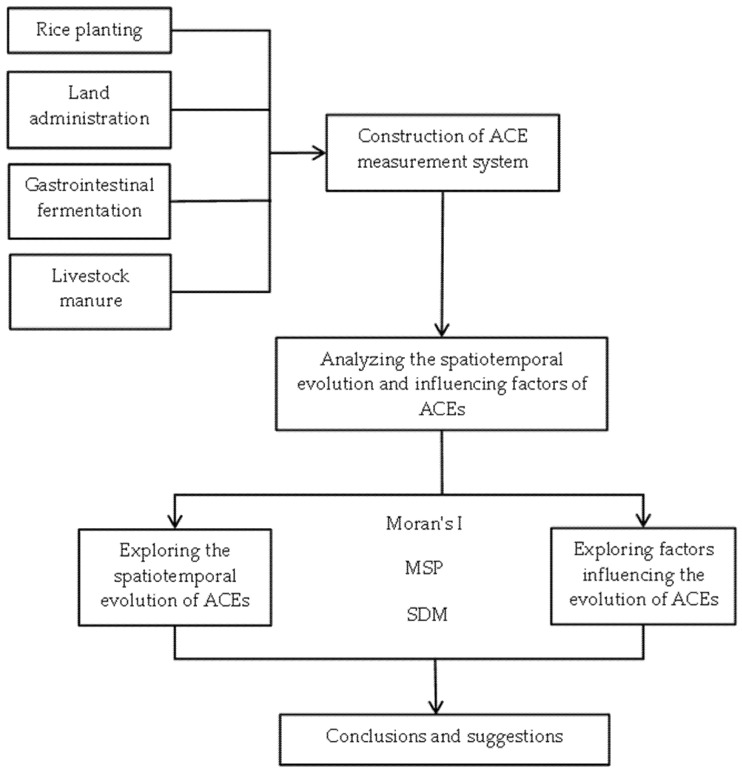
Analysis Framework.

**Figure 2 ijerph-20-00189-f002:**
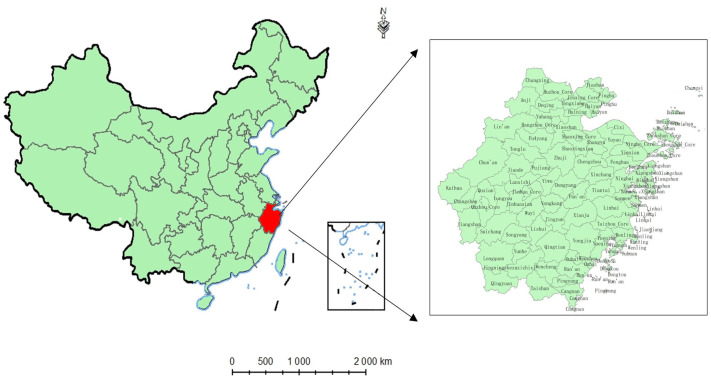
Geographical location and administrative division map of Zhejiang Province.

**Figure 3 ijerph-20-00189-f003:**
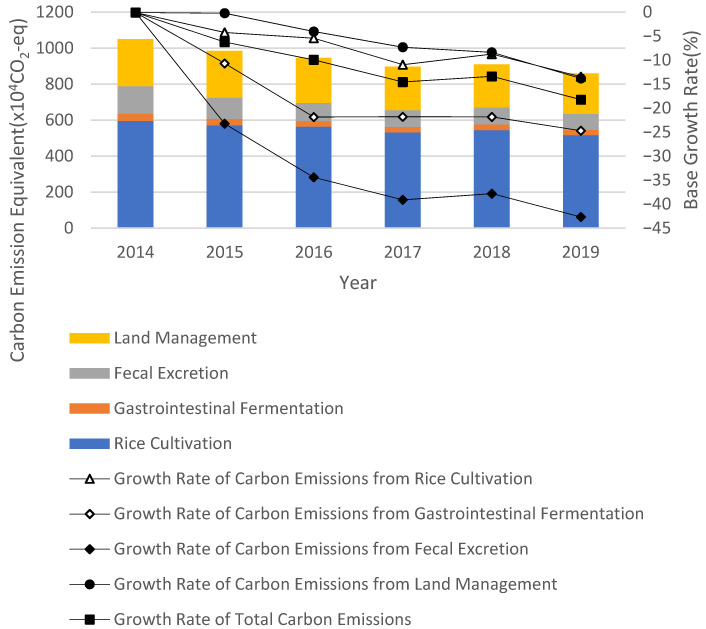
Change Trend of Agricultural Greenhouse Gas Emissions in Zhejiang Province from 2014 to 2019.

**Figure 4 ijerph-20-00189-f004:**
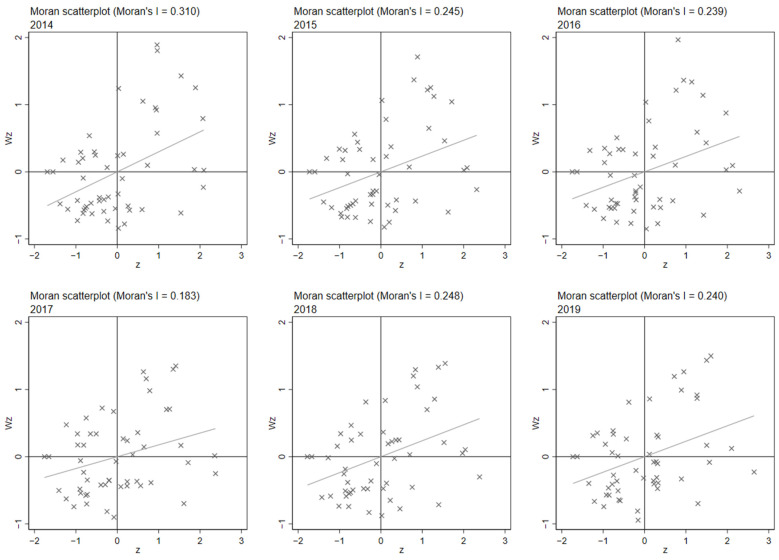
Scatter plot of Moran index.

**Table 1 ijerph-20-00189-t001:** Agricultural greenhouse gas emission sources and greenhouse gas emission coefficients.

Greenhouse Gas Emission Sources			Emission Coefficient	Data Sources
Rice planting	Early rice		14.37 g/m^2^	Wang Zhen et al. [38]
	Medium-season rice		57.96 g/m^2^	
	Late rice		34.5 g/m^2^	
Land administration	Agricultural machinery		0.18 kg·kW^−1^	Dubey [39]
	Chemical fertilizer	Nitrogenous fertilizer	857.54 kg·Mg^−1^	West and Marland [40]
		Phosphate fertilizer	165.09 kg·Mg^−1^	
		Potash fertilizer	120.28 kg·Mg^−1^	
		Compound fertilizer	380.97 kg·Mg^−1^	
	Pesticides		18.1 kg·kg^−1^	
	Agricultural film		19.0 kg·kg^−1^	ORNL [41]
	Ploughing		312.6 kg·hm^−2^	IABCAU [42]
	Irrigation		266.48 kg·hm^−2^	West and Marland [40]
Gastrointestinal fermentation (CH_4_)	Cattle		54.33 kg/(head·a)	IPCC [43]
	Pig		1 kg/(head·a)	
	Sheep		5 kg/(head·a)	
	Rabbit		0.254 kg/(head·a)	
Livestock manure (CH_4_ N_2_O)	Cattle (CH_4_)		5.33 kg/(head·a)	IPCC [43], FAO [44]
	Cattle (N_2_O)		1.24 kg/(head·a)	
	Pig (CH_4_)		3 kg/(head·a)	
	Pig (N_2_O)		0.53 kg/(head·a)	
	Sheep (CH_4_)		0.16 kg/(head·a)	
	Sheep (N_2_O)		0.33 kg/(head·a)	
	Rabbit (CH_4_)		0.08 kg/(head·a)	
	Rabbit (N_2_O)		0.02 kg/(head·a)	

**Table 2 ijerph-20-00189-t002:** Selection and descriptive statistics of variables affecting the ACESs in the county.

Variable	Mean	SD	Min	Max
Development level of the agricultural economy (AGDP) yuan (CNY)/person	3282.730	1825.931	23.838	8205.791
Urbanization rate (Urban) %	0.354	0.131	0.088	0.767
Rural population (R), 10,000 persons	49.438	31.513	4.010	130.360
Total power of agricultural machinery (AM) 10,000 kW	28.389	19.605	5.767	119.644
Disposable income of rural residents per capita (INC) yuan (CNY)	23,304.390	6835.956	11,739.000	39,529.000
Structure of the agricultural industry (Str) %	0.569	0.210	0.002	0.841
Crop planting structure (Cps) %	0.494	0.157	0.101	2.308
Agglomeration degree of the agricultural industry (IA) %	0.982	0.519	0.239	2.321
Human capital (Edu), 10,000 persons	2.597	1.671	0.187	7.669

**Table 3 ijerph-20-00189-t003:** Structure and Intensity of Agricultural Greenhouse Gas Emissions and Total Agricultural Greenhouse Gas Emissions in Zhejiang Province from 2014 to 2019.

Year	Rice Planting		Gastrointestinal Fermentation		Fecal Excretion		Land Administration		Total Amount * (10,000 tons)	Strength (ton/CNY 10,000)
	Emissions (10,000 tons)	Proportion (%)	Emissions (10,000 tons)	Proportion (%)	Emissions (10,000 tons)	Proportion (%)	Emissions (10,000 tons)	Proportion (%)		
2014	595.640	56.778	40.369	3.848	152.669	14.553	260.386	24.821	1049.063	1.242
2015	570.925	58.999	36.094	3.667	117.320	11.918	260.031	26.416	984.370	1.207
2016	563.821	59.614	31.583	3.339	100.278	10.603	250.109	26.444	945.791	1.181
2017	530.950	59.173	31.608	3.523	93.127	10.379	241.600	26.926	897.285	1.253
2018	544.112	59.818	31.596	3.474	95.093	10.454	238.809	26.254	909.609	1.337
2019	516.009	60.084	30.438	3.544	87.686	10.210	224.683	26.162	858.817	1.303

Note: * The total ACESs in Zhejiang are the sum of the 52 counties (cities).

**Table 4 ijerph-20-00189-t004:** Structure and Intensity of Agricultural Greenhouse Gas Emissions and Total Agricultural Greenhouse Gas Emissions in Zhejiang Province from 2014 to 2019.

County	Year 2014				Year 2019				Rate of Change I	Rate of Change II
	Total Amount (10,000 tons)	Rank	Strength (ton/CNY 10,000)	Rank	Total Amount (10,000 tons)	Rank	Strength (ton/CNY 10,000)	Rank		
Anji	20.576	21	0.899	40	17.684	24	1.032	36	−14.054	14.823
Cangnan	23.716	16	1.605	13	18.557	23	1.874	12	−21.752	16.768
Changshan	13.627	34	2.022	8	10.397	35	1.994	10	−23.701	−1.400
Chunan	17.624	26	0.663	49	10.164	36	0.444	50	−42.329	−33.086
Cixi	31.388	11	0.726	48	31.762	3	0.900	42	1.191	23.938
Daishan	1.786	51	0.952	38	1.045	51	0.948	40	−41.516	−0.422
Deqing	27.514	14	3.830	2	12.902	30	2.504	5	−53.108	−34.624
Dongyang	21.622	20	1.168	30	17.618	25	1.299	26	−18.518	11.164
Haining	31.645	8	1.783	11	23.352	13	2.005	9	−26.207	12.440
Haiyan	38.456	6	2.518	4	24.975	12	2.550	4	−35.056	1.276
Jiashan	31.577	9	1.067	35	25.529	10	1.322	25	−19.153	23.917
Jiande	20.301	24	0.783	44	18.629	22	0.773	47	−8.240	−1.292
Jiangshan	38.390	7	2.168	6	28.781	7	2.257	6	−25.029	4.112
Jinyun	10.684	40	1.204	25	8.222	43	1.128	35	−23.044	−6.302
Jingning	8.795	46	1.309	21	7.151	46	1.518	20	−18.694	15.971
Kaihua	15.093	31	1.300	22	10.907	32	1.296	27	−27.737	−0.244
Lanxi	28.836	13	1.480	17	19.183	20	1.245	32	−33.478	−15.863
Leqing	17.289	28	1.327	19	19.342	18	2.082	8	11.871	56.844
Linhai	27.282	15	0.928	39	24.987	11	0.967	38	−8.413	4.253
Longquan	20.510	22	1.585	15	15.034	27	1.519	19	−26.697	−4.161
Longyou	44.967	1	4.965	1	31.423	4	4.859	1	−30.120	−2.133
Ninghai	23.279	17	1.414	18	19.310	19	1.440	21	−17.050	1.809
Panan	3.824	50	0.330	52	3.710	50	0.389	52	−2.975	18.131
Pinghu	42.571	4	3.040	3	28.552	9	3.235	2	−32.932	6.426
Pingyang	20.409	23	2.168	5	19.506	16	2.636	3	−4.426	21.575
Pujiang	9.740	44	1.130	33	5.666	47	0.743	48	−41.833	−34.290
Qingtian	10.402	42	1.481	16	9.089	40	1.524	18	−12.618	2.945
Qingyuan	12.692	36	1.958	10	7.741	44	1.845	14	−39.008	−5.771
Ruian	16.339	30	1.321	20	18.679	21	1.850	13	14.317	40.032
Sanmen	9.087	45	1.010	37	9.270	39	1.253	31	2.018	23.962
Chengsi	0.168	52	1.170	29	0.069	52	0.847	45	−59.242	−27.639
Chengzhou	31.589	10	0.831	43	28.583	8	1.019	37	−9.516	22.686
Songyang	11.403	38	0.843	42	8.373	42	0.848	44	−26.575	0.543
Suichang	12.193	37	1.253	24	9.346	38	1.172	34	−23.349	−6.450
Taishun	11.046	39	1.673	12	10.419	34	1.714	16	−5.671	2.421
Tiantai	15.061	32	1.282	23	16.278	26	1.583	17	8.085	23.495
Tonglu	13.937	33	0.776	45	12.336	31	0.824	46	−11.485	6.224
Tongxiang	44.684	3	2.018	9	30.795	5	2.169	7	−31.082	7.445
Wenling	22.244	18	0.892	41	19.501	17	0.934	41	−12.333	4.667
Wencheng	8.778	47	1.139	31	9.371	37	1.352	23	6.760	18.730
Wuyi	17.466	27	1.190	26	14.919	28	1.271	30	−14.583	6.857
Xianju	16.385	29	1.180	27	14.547	29	1.226	33	−11.218	3.882
Xiangshan	21.932	19	1.131	32	19.639	14	1.285	29	−10.455	13.629
Xinchang	10.436	41	0.498	51	9.076	41	0.560	49	−13.031	12.483
Yiwu	10.308	43	0.525	50	7.511	45	0.424	51	−27.129	−19.151
Yongjia	19.641	25	2.156	7	19.633	15	1.936	11	−0.040	−10.169
Yongkang	12.953	35	1.588	14	10.630	33	1.807	15	−17.934	13.781
Yuyao	31.093	12	0.764	46	30.775	6	0.892	43	−1.025	16.830
Yuhuan	4.647	49	0.759	47	4.662	49	0.957	39	0.304	26.135
Yunhe	6.020	48	1.177	28	4.996	48	1.291	28	−17.009	9.665
Changxing	42.252	5	1.119	34	36.508	2	1.325	24	−13.596	18.401
Zhuji	44.802	2	1.060	36	41.683	1	1.359	22	−6.963	28.176

Note: Due to space limitations, Table 4 only lists the calculation results for each county in 2014 and 2019. The change rate I is the increase or decrease in agricultural greenhouse gas emissions in 2019 compared with those in 2014, and the change rate II is the increase or decrease in agricultural greenhouse gas emission intensity in 2019 compared with those in 2014.

**Table 5 ijerph-20-00189-t005:** Spatial autocorrelation analysis of the ACESs in counties based on global Moreland index.

Year	*W1*			*W2*			*W3*		
	Moran’s *I*	z	*p*	Moran’s *I*	z	*p*	Moran’s *I*	z	*p*
2014	0.310	6.151	0.002	0.105	2.850	0.003	0.267	5.121	0.002
2015	0.245	5.020	0.011	0.082	2.285	0.021	0.214	4.187	0.013
2016	0.239	4.733	0.013	0.076	2.241	0.022	0.207	4.052	0.017
2017	0.183	3.266	0.040	0.046	1.755	0.075	0.154	3.100	0.050
2018	0.248	4.052	0.010	0.062	2.310	0.040	0.198	3.886	0.019
2019	0.240	4.146	0.012	0.064	2.245	0.041	0.184	3.656	0.023

**Table 6 ijerph-20-00189-t006:** Regional distribution of Moran’s *I* during 2014–2019.

Year	HH Region	LH Region	LL Region	HL Region
2014	Anji, Cixi, Deqing, Haining, Haiyan, Jiashan, Jiande, Lanxi, Longyou, Pinghu, Shengzhou, Tongxiang, Xiangshan, Yuyao, Changxing	Changshan, Yueqing, Pujiang, Sanmen, Suichang, Tonglu, Yiwu, Yuhuan	Chun’an, Daishan, Jinyun, Jingning, Kaihua, Pan’an, Qingtian, Qingyuan, Ryan, Shengsi, Songyang, Taishun, Tiantai, Wencheng, Wuyi, Xianju, Xinchang, Yongjia, Yongkang, Yunhe	Cangnan, Dongyang, Jiangshan, Linhai, Longquan, Ninghai, Pingyang, Wenling, Zhuji
2015	Anji, Cixi, Deqing, Haining, Haiyan, Jiashan, Jiande, Lanxi, Longyou, Pinghu, Shengzhou, Tongxiang, Xiangshan, Yuyao, Changxing	Changshan, Yueqing, Pujiang, Sanmen, Suichang, Tonglu, Yiwu, Yuhuan	Chun’an, Daishan, Dongyang, Jinyun, Jingning, Kaihua, Pan’an, Pingyang, Qingtian, Qingyuan, Ryan, Shengsi, Songyang, Taishun, Tiantai, Wencheng, Wuyi, Xianju, Xinchang, Yongkang, Yunhe	Cangnan, Jiangshan, Linhai, Longquan, Ninghai, Wenling, Yongjia, Zhuji
2016	Anji, Cixi, Deqing, Haining, Haiyan, Jiashan, Jiande, Lanxi, Longyou, Pinghu, Shengzhou, Tongxiang, Xiangshan, Yuyao, Changxing	Changshan, Yueqing, Pujiang, Sanmen, Suichang, Tonglu, Yiwu, Yuhuan	Chun’an, Daishan, Dongyang, Jinyun, Jingning, Kaihua, Pan’an, Pingyang, Qingtian, Qingyuan, Ryan, Shengsi, Songyang, Taishun, Tiantai, Wencheng, Wuyi, Xianju, Xinchang, Yongkang, Yunhe	Cangnan, Jiangshan, Linhai, Longquan, Ninghai, Wenling, Yongjia, Zhuji
2017	Cixi, Haining, Haiyan, Jiashan, Jiande, Lanxi, Yueqing, Longyou, Pinghu, Pingyang, Shengzhou, Tongxiang, Xiangshan, Yuyao	Anji, Changshan, Deqing, Pujiang, Sanmen, Suichang, Tonglu, Yiwu, Yuhuan	Chun’an, Daishan, Dongyang, Jinyun, Jingning, Kaihua, Longquan, Pan’an, Qingtian, Qingyuan, Shengsi, Songyang, Taishun, Tiantai, Wencheng, Wuyi, Xianju, Xinchang, Yongkang, Yunhe	Cangnan, Jiangshan, Linhai, Ninghai, Ruian, Wenling, Yongjia, Changxing, Zhuji
2018	Anji, Cixi, Haining, Haiyan, Jiashan, Jiande, Lanxi, Yueqing, Longyou, Pinghu, Shengzhou, Tongxiang, Wenling, Xiangshan, Yuyao, Yuhuan, Changxing	Changshan, Deqing, Pujiang, Suichang, Tonglu, Yiwu	Chun’an, Daishan, Dongyang, Jinyun, Jingning, Kaihua, Pan’an, Qingtian, Qingyuan, Sanmen, Shengsi, Songyang, Taishun, Tiantai, Wencheng, Wuyi, Xianju, Xinchang, Yongkang, Yunhe	Cangnan, Jiangshan, Linhai, Longquan, Ninghai, Pingyang, Ruian, Yongjia, Zhuji
2019	Anji, Cixi, Dongyang, Haining, Haiyan, Jiashan, Yueqing, Pinghu, Shengzhou, Tongxiang, Xiangshan, Yuyao, Changxing	Changshan, Deqing, Pujiang, Sanmen, Suichang, Tonglu, Xinchang, Yiwu, Yuhuan	Chun’an, Daishan, Jinyun, Jingning, Kaihua, Longquan, Pan’an, Qingtian, Qingyuan, Shengsi, Songyang, Taishun, Tiantai, Wencheng, Wuyi, Xianju, Yongkang, Yunhe	Cangnan, Jiande, Jiangshan, Lanxi, Linhai, Longyou, Ninghai, Pingyang, Ruian, Wenling, Yongjia, Zhuji

**Table 7 ijerph-20-00189-t007:** Global Moran Index of the Agricultural Greenhouse Gas Emissions and Influencing Factors from 2014 to 2019.

YEAR	WSQT	AGDP	Urban	RP	AM	INC	Str	Cps	IA	Edu
2014	0.310 ***	0.364 ***	0.563 ***	0.437 ***	0.088	0.675 ***	0.354 ***	0.143 *	0.578 ***	0.288 ***
2015	0.245 **	0.346 ***	0.564 ***	0.430 ***	0.073	0.672 ***	0.364 ***	0.153 *	0.589 ***	0.300 ***
2016	0.239 **	0.336 ***	0.550 ***	0.394 ***	0.145 *	0.666 ***	0.354 ***	0.141 *	0.584 ***	0.311 ***
2017	0.183 **	0.300 ***	0.582 ***	0.399 ***	0.125 *	0.664 ***	0.359 ***	0.170 **	0.590 ***	0.311 ***
2018	0.248 **	0.306 ***	0.598 ***	0.401 ***	0.115 *	0.658 ***	0.364 ***	−0.107	0.583 ***	0.312 ***
2019	0.240 **	0.299 ***	0.586 ***	0.423 ***	0.077	0.653 ***	0.355 ***	0.196 **	0.590 ***	0.342 ***

***^,^ ** and * represent the significance levels of 1%, 5% and 10%, respectively.

**Table 8 ijerph-20-00189-t008:** Spatial measurement estimation results of the impacts of the variables on the ACESs under different spatial matrices.

Variable	Regression I *		Regression II		Regression III	
	Coefficient	Z-Value	Coefficient	Z-Value	Coefficient	Z-Value
lnAGDP	0.005	0.320	0.017	0.890	0.021	1.250
Urban	0.014	0.140	0.154	1.410	0.200 *	1.890
lnRP	0.245 **	2.150	0.306 **	2.270	0.326 ***	2.660
lnAM	−0.003	−0.050	0.069	0.880	0.061	0.840
lnINC	−1.576 ***	−10.030	−3.153 ***	−3.220	−3.426 ***	−3.240
Str	0.267	0.820	0.069	0.200	0.270	0.810
Cps	0.671 ***	16.090	0.682 ***	14.530	0.682 ***	15.840
IA	−0.168	−0.870	−0.160	−0.750	−0.224	−1.130
IA2	0.088 *	1.190	0.125 *	1.530	0.130 *	1.700
Edu	0.050 *	1.530	0.042	1.150	0.049	1.430
WlnAGDP	−0.003	−0.080	−0.178 *	−0.870	−0.059	−0.930
WUrban	−0.116	−0.910	−0.396	−1.450	−0.411 **	−2.430
WlnRP	0.085	0.450	−0.804	−0.670	0.040	0.100
WlnAM	0.126	1.140	−0.052	−0.100	0.155	0.770
WlnINC	1.454 ***	8.450	2.654 **	2.400	3.341 ***	3.060
WStr	−1.315 ***	−2.670	−5.435 ***	−3.610	−1.709 **	−2.400
WCps	−0.156 *	−1.650	−0.118	−0.330	−0.191 *	−1.710
WIA	0.244	0.770	0.949	0.610	1.022 *	1.720
WIA2	0.024	0.180	0.542	0.980	−0.139	−0.600
WEdu	0.039	0.670	0.316	1.190	0.057	0.550
*ρ*	0.373 ***	6.070	0.573 ***	3.800	0.572 ***	6.240
sigma2_e	0.006 ***	12.220	0.008 ***	12.400	0.007 ***	12.280

* Regression I, regression II and regression III, respectively, correspond to the regression results of the fixed effect space Dubin model for the adjacent space weight matrix, inverse distance space weight matrix and inverse distance square weight matrix. ***^,^ ** and * represent the significance levels of 1%, 5% and 10%, respectively.

**Table 9 ijerph-20-00189-t009:** Direct and indirect effects of the various factors on agricultural carbon emissions.

Variable	Direct Effect		Indirect Effect		Total Effect	
	Coefficient	Z-Value	Coefficient	Z-Value	Coefficient	Z-Value
lnAGDP	0.005	0.290	−0.004	−0.090	0.001	0.010
Urban	−0.003	−0.040	−0.142	−0.910	−0.145	−0.950
RP	0.280 **	2.460	0.234	0.920	0.514 *	1.680
AM	0.014	0.200	0.179	1.210	0.193	1.080
lnINC	−1.456 ***	−10.100	1.205 ***	7.360	−0.251 **	−2.000
Str	0.124	0.400	−1.692 ***	−2.800	1.568 **	−2.390
Cps	0.684 ***	13.810	0.138	1.150	0.822 ***	5.320
IA	−0.153	−0.760	0.221	0.470	0.068	0.120
IA^2^	0.101 *	1.320	0.091	0.490	0.191	0.860
Edu	0.059 *	1.740	0.086	1.080	0.145	1.540

***^,^ ** and * represent the significance levels of 1%, 5% and 10%, respectively.

**Table 10 ijerph-20-00189-t010:** Spatial metrological estimation results of the impacts of various variables on the ACESs of the regional samples.

Variable	The 26 Counties in Mountainous Areas		Counties in Non-Mountainous Areas	
	Coefficient	Z-Value	Coefficient	Z-Value
lnAGDP	−0.001	0.020	0.032	0.780
Urban	0.009	0.060	−0.025	−0.160
lnRP	0.084	0.680	0.520 *	1.870
lnAM	−0.017	−0.210	−0.018	−0.150
lnINC	0.228	0.160	−0.526 ***	−3.340
Str	0.273	0.800	1.174 *	1.730
Cps	0.619 ***	2.790	0.653 ***	11.790
IA	0.117	0.470	−0.056	−0.120
IA2	0.013	0.170	0.071	0.280
Edu	0.139 ***	3.580	−0.019	−0.310
WlnAGDP	0.040	1.290	0.381	1.620
WUrban	−0.393 **	−2.080	0.004	0.020
WlnRP	−0.472 ***	−3.030	0.293	0.620
WlnAM	−0.271 **	−2.100	0.106	0.610
WlnINC	−0.388	−0.280	0.661 ***	2.580
WStr	−1.199 **	−2.150	−2.944 ***	−3.830
WCps	1.098 ***	3.040	−0.056	−0.440
WIA	0.424	0.090	0.316	0.480
WIA2	0.033	0.220	−0.123	−0.280
WEdu	−0.029	−0.450	0.135	1.170
*ρ*	0.443 ***	5.400	0.201 ***	2.150
sigma2_e	0.003 ***	8.080	0.010 ***	9.230

Note: ‘*’, ‘**’, ‘***’ represent the significance levels of 10%, 5% and 1%, respectively.

**Table 11 ijerph-20-00189-t011:** Estimation of the direct and indirect effects of various factors on the ACESs by region (26 counties in the mountainous area).

Variable	Direct Effect		Indirect Effect		Total Effect	
	Coefficient	Z-Value	Coefficient	Z-Value	Coefficient	Z-Value
lnAGDP	0.008	0.470	0.061	1.220	0.068	1.160
Urban	−0.062	−0.440	−0.614 **	−2.510	−0.677 ***	−2.970
lnRP	0.0189	0.160	−0.718 ***	−2.840	−0.699 **	−2.360
lnAM	−0.065	−0.710	−0.448 *	−1.890	−0.513 *	−1.700
lnINC	0.145	0.120	−0.435	−0.370	0.290 **	−2.010
Str	0.106	0.290	−1.735 *	−1.880	1.629	−1.420
Cps	0.857 ***	4.250	2.254 ***	4.120	3.111 ***	5.870
IA	0.179	0.610	0.706	0.840	0.886	0.840
IA2	0.027	0.290	0.085	0.320	0.112	0.340
Edu	0.146 ***	3.290	0.063	0.570	0.209	1.500

Note: ‘*’, ‘**’, ‘***’ represent the significance levels of 10%, 5% and 1%, respectively.

**Table 12 ijerph-20-00189-t012:** Estimation of the direct and indirect effects of various factors on the ACESs by region (non-mountainous counties).

Variable	Direct Effect		Indirect Effect		Total Effect	
	Coefficient	Z-Value	Coefficient	Z-Value	Coefficient	Z-Value
lnAGDP	0.063	1.130	0.370 *	1.680	0.433 *	1.790
Urban	−0.029	−0.210	0.014	0.080	−0.015	−0.080
lnRP	0.581 **	2.150	0.364	0.800	0.944 *	1.690
lnAM	−0.007	−0.060	0.107	0.650	0.100	0.520
lnINC	−0.476 ***	−3.150	0.536 **	2.290	0.060	0.250
Str	1.001	1.570	−2.718 ***	−4.020	−1.717 **	−2.310
Cps	0.660 ***	10.600	0.080	0.710	0.739 ***	4.830
IA	−0.043	−0.100	0.258	0.380	0.216	0.240
IA^2^	0.076	0.300	−0.085	−0.190	−0.010	−0.020
Edu	−0.006	−0.110	0.134	1.200	0.128	1.120

Note: ‘*’, ‘**’, ‘***’ represent the significance levels of 10%, 5% and 1%, respectively.

## Data Availability

The datasets used and analyzed during the current study are available from the corresponding author on reasonable request.

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
