# Peer review of "Study on the Spatiotemporal Evolution and Influencing Factors of Agricultural Carbon Emissions in the Counties of Zhejiang Province"

_ijerph, 2022, doi:10.3390/ijerph20010189_

Round 1
Reviewer 1 Report
The paper considers an important and timely topic, emissions of agriculture. Agriculture is a significant source of both CO2 and CH4, and those emissions must be reduced in order to mitigate climate change. The paper provides interesting insights, but some modifications are needed.
The abstract should be a total of about 200 words maximum. Your abstract is almost 500 words, please squeeze.
Tables: Please capitalize the initials and use superscript in “m2” (Table 1). In the tables, please ensure that the brackets are in the same line.
In page 6 (line 225->), an equation is embedded to text. For clarity, please locate the equation separately as in cases of previous equations. Please use italics in equations and variables consistently.
Line 236-238, please add a reference for the used data. Also, in lines 353-354 and 450-451, add a reference when you talk about a previous article.
For me, it seems that methods and results sections are more or less mixed. In my opinion, all methods and used data should be presented in the methods section and there should be a separate section for results.
I feel that the research period (2014-2019) is rather short. I understand it may require too much work to extend the research period, but it would be nice if you could include some long-term views, i.e., how the emissions are developed within previous decades.
Author Response
First, we would like express our deep gratitude for your comments. They are very helpful and have greatly improved the quality of the manuscript.
Point 1: The abstract should be a total of about 200 words maximum. Your abstract is almost 500 words, please squeeze.
Response 1: We rewrote the summary and reduced the number of words in the summary.
Point 1: Tables: Please capitalize the initials and use superscript in “m2” (Table 1). In the tables, please ensure that the brackets are in the same line.
Response 1:All tables are capitalized and superscripts are used in "m2" (Table 1). In the table, ensure that the brackets are on the same line.
Point 3: In page 6 (line 225->), an equation is embedded to text. For clarity, please locate the equation separately as in cases of previous equations. Please use italics in equations and variables consistently.
Response 3:The equations and variables in the article are changed to italics.
Point 4: Line 236-238, please add a reference for the used data. Also, in lines 353-354 and 450-451, add a reference when you talk about a previous article.
Response 4:In lines 236-238 of the original manuscript, in lines 353-354 and 450-451 ,add the reference source of the data used.
Point 5: For me, it seems that methods and results sections are more or less mixed. In my opinion, all methods and used data should be presented in the methods section and there should be a separate section for results.
Response5:Following your suggestions, we adjusted the structure of the full text, and showed the research methods and results separately.The overall structure of the paper has been readjusted to 1. Introduction; 2. Materials and Methods; 3. Results; 4. discussion; 5. Conventions; 6. Suggestions.
Point 6:I feel that the research period (2014-2019) is rather short. I understand it may require too much work to extend the research period, but it would be nice if you could include some long-term views, i.e., how the emissions are developed within previous decades.
Response6: Since there is no detailed data for measuring agricultural carbon sources in the relevant public statistical yearbooks, the original agricultural carbon emissions data and rural population data used in this paper are derived from the agricultural statistics of Zhejiang Province from 2014 to 2019. For example, in the open statistical yearbook, the area of rice is only the total amount, while in the agricultural statistics of Zhejiang Province, the area of early rice, middle rice and late rice is recorded in detail, which is conducive to improving the accuracy of carbon emissions in this study. As the agricultural statistics of Zhejiang Province refer to the internal data of the Department of Agriculture and Rural Affairs of Zhejiang Province, only the paper version from 2014 to 2019 can be obtained at present. We manually input and collate the paper version. If the later data are available, we will follow your suggestions and conduct long-term research on agricultural carbon emissions.
Reviewer 2 Report
This manuscript (ijerph-2099945) tries to analyze the spatial and temporal evolution and influencing factors of agricultural carbon emissions in the county of Zhejiang Province, China. Although the study fits the aim and scope of this journal and the amount of the work is enough, its novelty and contribution to agricultural carbon emissions research needs to be highlighted throughout the manuscript. Another serious concern is that some related latest studies have been neglected. Also, the current results of this study can hardly be reviewed because of those problems about data and methodology. Therefore, a “Major Revision” is required. My detailed suggestions and comments are presented as follows:
- 1. First of all, the overall structure of the paper needs to be deepened and the significance and innovation of the research needs to be further demonstrated.
- 2. The authors need to explain more clearly why the "county" was selected as the basic research unit. The authors mentioned that: "County is the main battlefield of agricultural production", but what about the upper and lower administrative units?
- 3. The scientific question or research gap is missing in the Abstract. The novelty / originality should be clearly justified that the manuscript contains sufficient contributions to the new body of knowledge from the international perspective. The authors just mentioned what they have done in a specific study area, but without explaining why they have to do so in this way. Similarly, the introduction section is weak because the authors failed to raise a fundamental scientific question or gap beyond this study area. Therefore, potential readers can hardly identify the need that the authors should have to provide a new solution from an international perspective. Note that the analysis of the spatial and temporal evolution and influencing factors of agricultural carbon emissions is not a new attempt in carbon emissions research (see below for example). For example, what is the real difference between the Zhejiang and Jiangsu Province?
Spatial and Temporal Characteristics and Drivers of Agricultural Carbon Emissions in Jiangsu Province, China. Int. J. Environ. Res. Public Health, 2022, 19 (19): 12463
- 4. Actually, there will never has a unified conclusion on the influencing factors of agricultural carbon emissions because the influencing factors will definitely differ greatly across different study areas.
- 5. Table 1: please clearly explain how to determine the proper coefficients for these different carbon sources adapted to the study area, Zhejiang Province. The references listed in this table were analyzing over ten years ago, and were not directly related to this study area.
- 6. The social factors affect agricultural greenhouse gas emissions from a micro level are also very important to be considered.
- 7. A further and detailed literature review must be conducted. The Spatial Durbin model is not the current state-of-the-art method for analyzing mathematical relationships because carbon emissions are non-linear systems. Many non-linear methods, such as the random forest, have already been utilized for analyzing the drivers of carbon emissions during the past decade (see below for example). These methods should be compared and discussed in details.
Analyzing the impact of three-dimensional building structure on CO2 emissions based on random forest regression. Energy, 2021, 236: 121502
Analyzing the Impact of Urbanization and Energy-Related Factors upon CO2 Emissions in Central–Eastern European Countries by Using Machine Learning Algorithms and Panel Data Analysis. Energies, 2021
- 8. The entire analysis framework should be presented in one new figure.
- 9. Section 2.2 Variable selection: please clearly explain why these and why just these factors have been considered in this study.
- 10. Section 3.2. Analysis of inter county differences in agricultural carbon emissions: the authors need to present the location of those different counties.
- 11. The changes of the global Moran Index of Agricultural Greenhouse Gas Emissions and Influencing Factors from 2014 to 2019 were indeed very trivial.
- 12. From Table 5, we can see that there is no specific or regular rule for the changing in many factors (sometimes increasing, and sometimes decreasing).
- 13. Takeaway for practice is also encouraged to be included in this manuscript. It should be clear enough to present your implications and recommendations for both local and international practice.
Author Response
Response to the Second Reviewer Comments
First, we would like express our deep gratitude for your comments. They are very helpful and have greatly improved the quality of the manuscript.
Point 1: First of all, the overall structure of the paper needs to be deepened and the significance and innovation of the research needs to be further demonstrated..
Response 1: The overall structure of the paper has been readjusted to 1. Introduction; 2. Materials and Methods; 3. Results; 4. discussion; 5. Conventions; 6. Suggestions. 2.1. Analytical Framework is mainly added, which divides the original conclusion into conclusions and recommendations. The introduction adds the description of the significance and innovation of the research.
Point 2: The authors need to explain more clearly why the "county" was selected as the basic research unit. The authors mentioned that: "County is the main battlefield of agricultural production", but what about the upper and lower administrative units?
Response 2: In the introduction, the reason why the county is selected as the research unit is added. Compared with the county data, the errors in provincial or municipal data are larger, while township-level data are difficult to obtain. The county is a basic regional unit with a relatively independent form of administration and relatively complete regionalism and comprehensiveness. Each county has relatively consistent natural conditions and a consistent social, economic and cultural background, which often feature among China's current statistical data. Therefore, it is more feasible to use the county as the vehicle of agricultural carbon emission reduction.
Point 3. The scientific question or research gap is missing in the Abstract. The novelty / originality should be clearly justified that the manuscript contains sufficient contributions to the new body of knowledge from the international perspective. The authors just mentioned what they have done in a specific study area, but without explaining why they have to do so in this way. Similarly, the introduction section is weak because the authors failed to raise a fundamental scientific question or gap beyond this study area. Therefore, potential readers can hardly identify the need that the authors should have to provide a new solution from an international perspective. Note that the analysis of the spatial and temporal evolution and influencing factors of agricultural carbon emissions is not a new attempt in carbon emissions research (see below for example). For example, what is the real difference between the Zhejiang and Jiangsu Province?
Response 2: According to your suggestion, we rewrote the summary, and added the explanation of scientific problems and marginal contribution of this paper in the introduction. the accurate measurement of agricultural carbon emissions and the analysis of the key influential factors and spatial effects are the premise of the rational formulation of agricultural emission reduction policies and the promotion of the regional coordinated governance of reductions in agricultural carbon emissions.
Zhejiang Province and Jiangsu Province are located in the eastern coastal zone of China, with a large difference in agricultural natural endowment. Comparatively speaking, Jiangsu's terrain is dominated by plains, which account for about 69% of the total land area of the province. Zhejiang Province has "seven mountains, one water and two fields". The mountainous area accounts for the vast majority of the total area, and the plain area is much smaller than that of Jiangsu Province. The difference of agricultural natural endowment has led to a large difference between the two provinces in terms of planting structure, agricultural development mode, etc. These differences will affect the spatial correlation and main driving factors of agricultural carbon emissions. Because of these differences, we need to strengthen the research on regional agricultural carbon emissions Int. J. Environ. Res. Of course, the existing models and methods for various factors affecting agricultural carbon emissions have their advantages and disadvantages. For example, global climate change, as a cross cutting issue between nature and society, involves a large number of impact factors, but the increase in the number of indicators is likely to cause a dimension disaster for analysis. Machine learning has an inherent advantage in overcoming the dimension disaster through the information mining of a large amount of data, but the reliability of machine learning often requires sufficient samples. When the number of samples is small, Its advantages are also difficult to reflect.
Point 4. Actually, there will never has a unified conclusion on the influencing factors of agricultural carbon emissions because the influencing factors will definitely differ greatly across different study areas.
Response 4: yes,there will never has a unified conclusion on the influencing factors of agricultural carbon emissions because the influencing factors will definitely differ greatly across different study areas.The influencing factors of ACESs are very complex. It involves natural conditions, agricultural management methods and social factors. Different from natural conditions and agricultural management, social factors affect agricultural greenhouse gas emissions from a macro level. For this article, based on the existing literature about the analysis of the influencing factors of ACEs[24-29,32] and the availability of data, this paper selects the influencing factors from the perspective of technical economic and social perspective.
Point 5: Table 1: please clearly explain how to determine the proper coefficients for these different carbon sources adapted to the study area, Zhejiang Province. The references listed in this table were analyzing over ten years ago, and were not directly related to this study area.
Response 5: Our paper refers to the conclusions of natural science research, and the carbon emission coefficient of rice planting conforms to the local coefficient of Zhejiang. Rice planting is divided into three different coefficients, early rice, middle rice and late rice, to calculate carbon emissions respectively, and then add them up. The carbon emission coefficient of other carbon sources has been referred to for a long time, but it is still the most widely used and authoritative emission coefficient at present. Later, we will follow your suggestions and try to refer to the latest carbon emission coefficient of carbon sources.
Point 6:-The social factors affect agricultural greenhouse gas emissions from a micro level are also very important to be considered.
Response 6 :Following your suggestions, we have added three variables and classified the original variables into three categories: technical, economic and social.
Based on the existing literature regarding the analysis of the factors influencing ACEs [24-29,32] and the availability of data, this paper selects the influencing factors from the technical, economic and social perspectives, in which the technical factors are expressed in the total power of the agricultural machinery (AM), and the economic factors include the developmental level of the agricultural economy (AGDP), structure of the agricultural industry (Str), crop planting structure (Cps) and agglomeration degree of the agricultural industry (IA). The social factors include the urbanization rate (Urban), rural population (RP), disposable income of the rural residents per capita (INC), total power of the agricultural machinery (AM) and human capital (Edu).
Point 7: A further and detailed literature review must be conducted. The Spatial Durbin model is not the current state-of-the-art method for analyzing mathematical relationships because carbon emissions are non-linear systems. Many non-linear methods, such as the random forest, have already been utilized for analyzing the drivers of carbon emissions during the past decade (see below for example). These methods should be compared and discussed in details.
Response 7: We have added recent literature and compared the advantages and disadvantages of several types of methods in the introduction. Each method has its advantages and disadvantages. We choose the spatial econometric model based on the existing data constraints and the goal of studying spatial spillover effects. We will try the new method you recommend in the follow-up study.
The first type of research uses factor decomposition method to investigate the driving factors of agricultural carbon emissions. The factorization method includes SDA model, Laspeyres statistical index, Kaya identity, LDA model, Shapley algorithm, IPAT model, STIRPAT model, etc.[31]. The factorization method can analyze the structure of complex technology-economy-social factors. In particular, the STIRPAT model can be expanded according to agricultural production practices in the study area, so it is widely used [32]. However, the meaning of factors decomposed by factor decomposition analysis to satisfy the identity relationship is weakened, and even the interpretation is one-sided. The second type of machine learning has inherent advantages in overcoming the disaster of dimension through the information mining of large amounts of data. For example, the random forest algorithm has the characteristics of robustness against noise and outliers and stability of prediction, so it has been applied to the research on influencing factors of carbon emissions[33,34]. However, the random forest method will cause the problem of over-fitting due to the over-complexity of the training stage, and there are certain requirements on the number of input variables[33]. The third type of research uses typical econometric analysis methods as research tools, and general econometric methods include OLS, DID, etc., whose advantage is that it can explore the effect (positive or negative) and mechanism of independent variables on agricultural carbon emissions[8,25]. However, the traditional measurement method assumes that the agricultural carbon effect is independent between regions, which is inconsistent with the specific agricultural production practice and weakens the practical significance of the research conclusions.
Point 8: The entire analysis framework should be presented in one new figure.
Response 8: Under "2. Materials and Methods", 2.1. Analytical Framework and Figure 1. Analysis Framework are added.
Point 9: Section 2.2 Variable selection: please clearly explain why these and why just these factors have been considered in this study.
Response 9: With reference to existing research literature and our research purposes, we selected relevant variables in social, economic and technical aspects from a macro perspective. Of course, as you said, "there will never have a unified inclusion on the influencing factors of agricultural carbon emissions because the influencing factors will definitively different greatly across different study areas."
Point 10: Section 3.2. Analysis of inter county differences in agricultural carbon emissions: the authors need to present the location of those different counties.
Response 10 :Added “Figure 2. Geographic location and administrative division map of Zhejiang Province.”
Point 11: The changes of the global Moran Index of Agricultural Greenhouse Gas Emissions and Influencing Factors from 2014 to 2019 were indeed very trivial.
Response 11: We use three spatial weights to calculate the global Moran index, which shows that the spatial autocorrelation of Moran index calculated based on the adjacent weight matrix is the largest and most significant. The test with a significance level of 5% shows that the spatial effect of agricultural greenhouse gas emissions in the county is very obvious. In view of the fact that the global Moran index cannot show the local spatial agglomeration and local spatial autocorrelation characteristics of agricultural greenhouse gas emissions in the county, this paper also adds a Moran Scatter Map (MSP) to further reveal the local spatial characteristics of agricultural greenhouse gas emissions in the county. The results show that the spatial agglomeration is obvious.
Point- 12. From Table 5, we can see that there is no specific or regular rule for the changing in many factors (sometimes increasing, and sometimes decreasing).
Response 12: In view of our short research period, Moran index does not show obvious laws. We re run the software, and the Moran index test under various weight cases shows that it has passed the significance test. In this paper, the Moran index is mainly used to observe the spatial agglomeration effect of carbon emissions, and to make a basis for whether it is applicable to the spatial measurement model. Our analysis results show that the spatial correlation of agricultural carbon emissions is significant, and the agglomeration effect is significant.
Point13: Takeaway for practice is also encouraged to be included in this manuscript. It should be clear enough to present your implications and recommendations for both local and international practice.
Response 13: Following your suggestions, we have strengthened our contribution to the article in the introduction part and put forward suggestions for local practice in the suggestion part.
Round 2
Reviewer 1 Report
The authors covered given comments and suggestions rather nicely in the revised version. In my opinion, the paper can be published.
Reviewer 2 Report
Thank you for incorporating my comments and suggestions.